# Angular velocity integration in a fly heading circuit

Daniel Turner-Evans[1†], Stephanie Wegener[1†], Hervé Rouault[1], Romain Franconville[1], Tanya Wolff[1], Johannes D Seelig[2], Shaul Druckmann[1], Vivek Jayaraman[1*]

[1]Janelia Research Campus, Howard Hughes Medical Institute, Ashburn, United States; [2]Center of Advanced European Studies and Research (CAESAR), Bonn, Germany

**Abstract** Many animals maintain an internal representation of their heading as they move through their surroundings. Such a compass representation was recently discovered in a neural population in the *Drosophila melanogaster* central complex, a brain region implicated in spatial navigation. Here, we use two-photon calcium imaging and electrophysiology in head-fixed walking flies to identify a different neural population that conjunctively encodes heading and angular velocity, and is excited selectively by turns in either the clockwise or counterclockwise direction. We show how these mirror-symmetric turn responses combine with the neurons' connectivity to the compass neurons to create an elegant mechanism for updating the fly's heading representation when the animal turns in darkness. This mechanism, which employs recurrent loops with an angular shift, bears a resemblance to those proposed in theoretical models for rodent head direction cells. Our results provide a striking example of structure matching function for a broadly relevant computation.

*For correspondence: vivek@ janelia.hhmi.org

†These authors contributed equally to this work

Competing interests: The authors declare that no competing interests exist.

## Introduction

When navigating an environment, animals rely on a range of sensory cues from their surroundings to determine their actions. However, even in the absence of such external input, several animals, including insects, can rely on self-motion cues to maintain and update their bearings. In rodents, this ability is thought to rely on head direction cells, which represent the animal's orientation with respect to a fixed external landmark (*Taube et al., 1990*) and use vestibular signals to maintain their directional tuning in darkness (*Taube, 2007*). A variety of ring attractor models — theorized networks of neurons schematized as being arranged in a ring based on their directional tuning, with connectivity strengths depending on their mutual distances — have been proposed to explain how the brain might maintain and update such an internal compass representation (*Knierim and Zhang, 2012*; *Skaggs et al., 1995*; *Xie et al., 2002*). Neural activity in these structures is localized into a 'bump' comprised of co-active neurons with similar heading preferences. This activity bump, which represents the animal's angular orientation, moves around the ring as the animal changes its heading. While some models have proposed specific network configurations that would enable angular velocity signals to update this compass representation in darkness, for example, (*Skaggs et al., 1995*), experimental support for such mechanisms has been limited.

In the central brain of *Drosophila melanogaster*, a population of neurons has been found to track a tethered walking fly's virtual heading in visual surroundings and in darkness (*Seelig and Jayaraman, 2015*). These neurons reside in the central complex, a brain region comprised of several distinct neuropiles, among them the donut-shaped ellipsoid body and the handle-bar-shaped protocerebral bridge (*Figure 1A*). We will refer to these 'compass neurons' (PB$_{G1-8}$.b-EBw.s-D/Vgall.b [*Wolff et al.,*

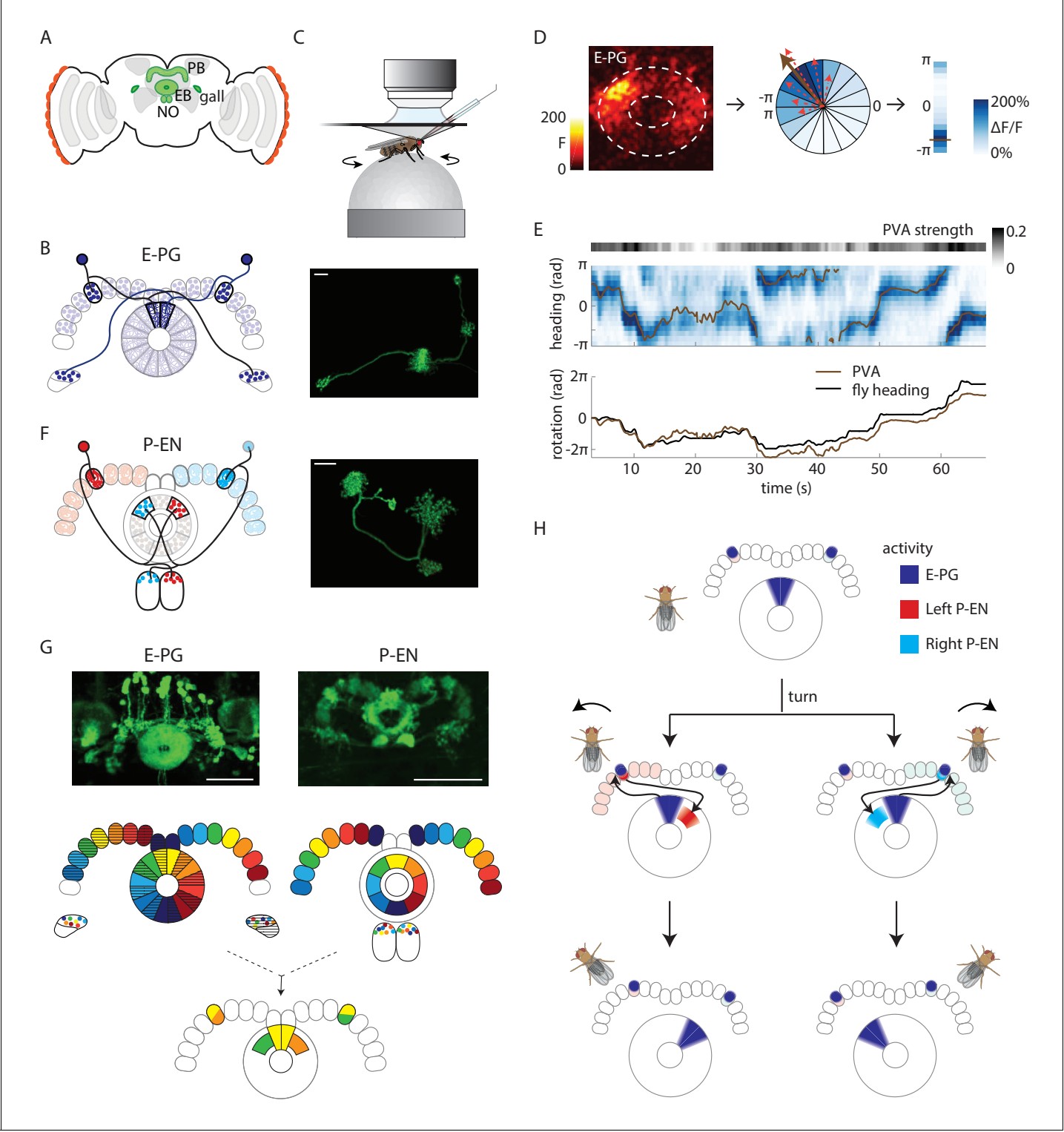

**Figure 1.** Anatomy suggests a potential circuit mechanism to update a compass representation. (**A**) Schematic of the fly brain. Highlighted in green are the ellipsoid body (EB), protocerebral bridge (PB), and the paired gall and noduli (NO). (**B**) (Left) Schematic of the morphology of two E-PG neurons innervating different sides of the protocerebral bridge. The probable direction of information flow is from their predominantly spiny arbors in the ellipsoid body ('E-') to their predominantly bouton-like projections in the protocerebral bridge and gall ('-PG'). (Right) A single GFP-labeled E-PG neuron. Scale bar: 10 μm. (**C**) Schematic of the head-fixed walking fly preparation used for two-photon calcium imaging and single cell electrophysiology. (**D**) For a given two-photon imaged volume (Left, maximum intensity projection of the ellipsoid body), the population vector average (PVA, brown arrow, Center) for a given time point is computed by summing vectors representing the instantaneous calcium activity of each sector of the

*Figure 1 continued on next page*

Figure 1 continued

ellipsoid body (for example, the dotted red arrows shown here for a few sectors). Each sector is a 22.5° slice of the ellipsoid body, defined manually, and each sector's vector points radially outward along its half angle. PVA strength is the normalized amplitude of the summed vector. (E) (Top) E-PG calcium activity (blue) in the ellipsoid body as the fly turns in darkness. The 16 ellipsoid body sectors are shown unwrapped from $-\pi$ to $\pi$. The PVA is shown in brown, PVA strength is at the top. (Bottom) Comparison of the fly's heading (black) with the PVA shows a tight correlation of the two, albeit with some drift. This example shows a trial in which the PVA closely matches the heading. We also observed larger, low frequency shifts between the two, as reported previously (*Seelig and Jayaraman, 2015*). (F) (Left) Schematic of the morphology of two P-EN neurons innervating different sides of the protocerebral bridge. P-EN neurons arborizing in the same protocerebral bridge glomeruli as their E-PG counterparts send processes to offset (neighboring) sectors of the ellipsoid body. Processes in the protocerebral bridge are overwhelmingly spiny and likely dendritic (*Wolff et al., 2015*). Processes in the ellipsoid body and noduli are predominantly bouton-like and suggestive of presynaptic specializations. (Right) A single GFP labeled P-EN neuron. Scale bar: 10 μm. (G) (Top left) GFP-labeled E-PG neurons in the R60D05 Gal4 line (maximum intensity projection, reproduced with permission from Janelia FlyLight Image Database (*Jenett et al., 2012*). (Top right) GFP-labeled P-EN neurons in the R37F06 Gal4 line (maximum intensity projection reproduced from Janelia FlyLight Image Database [*Jenett et al., 2012*]). (Middle left) Ellipsoid body to protocerebral bridge connectivity map for E-PG neurons. Single neurons that arborize in one wedge of the ellipsoid body arborize in the glomerulus of the same color and shading in the bridge. Arborizations in the gall do not exhibit a stereotyped pattern. (Middle right). Protocerebral bridge to ellipsoid body connectivity map for P-EN neurons. Single neurons that arborize in one glomerulus in the bridge arborize in the ellipsoid body tile with the corresponding color. Arborizations in the noduli do not exhibit stereotyped patterns. (Bottom) Single E-PG and P-EN neurons that arborize in the same protocerebral bridge glomerulus have non-overlapping processes in the ellipsoid body with a stereotyped angular shift between the two. All scale bars are 50 μm. (H) Overview of an anatomically motivated mechanism to update a heading representation. E-PG neurons are assumed to make excitatory connections onto P-EN neurons in the protocerebral bridge. P-EN neurons, in turn, are assumed to make excitatory connections onto E-PG neurons in the ellipsoid body. A bump of activity in E-PG neurons (dark blue) represents the fly's heading in the ellipsoid body. This bump of activity would result in two bumps of E-PG activity in the protocerebral bridge, one on either side. If the two sides of the bridge were to receive asymmetric input dependent on the fly's angular velocity and turning direction, P-EN neurons with dendrites in the bridge columns that also receive E-PG input would be activated. The anatomical shift between the P-EN and E-PG neurons in the ellipsoid body (compare B and F) would then cause the P-EN neurons to excite E-PG neurons nearby, shifting the E-PG activity bump and updating the heading representation. Note also that the mirror-symmetric activation of the two sides of the bridge would also be visible in activity differences in the noduli (see F and *Figure 2A*), each of which only receives P-EN projections from the opposite side of the brain.

*2015*]) as E-PG neurons, to signify their predominantly spiny (and, thus, putatively post-synaptic) projections within the ellipsoid body ('E-') and their predominantly bouton-like projections within the protocerebral bridge ('-P') and the gall ('G') (*Figure 1B*). Dendritic calcium activity in the E-PG population localizes into a single bump that moves around the ellipsoid body as the fly turns (*Figure 1C–E*), (*Seelig and Jayaraman, 2015*). In this study, we demonstrate how a distinctive recurrent circuit motif enables angular velocity signals carried by a different population of neurons to move the bump of E-PG population activity around the ellipsoid body so that it tracks the fly's angular orientation in darkness.

For our investigation of potential inputs that might move the E-PG bump, we focused on a population of neurons called $PB_{G2-9}.s\text{-}EBt.b\text{-}NO_1.b$ neurons (*Wolff et al., 2015*), P-EN neurons for short, to signify their spiny arbors in the protocerebral bridge and their predominantly bouton-like — and therefore likely presynaptic — projections within the ellipsoid body and noduli (see *Figure 1F*). The anatomy and polarity of P-EN neurons relative to E-PG neurons suggested a possible mechanism for how these neurons might update the position of an existing E-PG bump in the ellipsoid body: Each E-PG neuron putatively relays information from one of 16 wedge-shaped slices of the ellipsoid body to a single protocerebral bridge slice (*Ito et al., 2014*), also known as a glomerulus (*Wolff et al., 2015*) (*Figure 1B*). In contrast, the morphology of P-EN neurons suggests that they each have dendrites in one of the bridge glomeruli and send outputs to one of 8 tile-shaped sectors of the ellipsoid body (*Figure 1F*). Thus, a single P-EN tile overlaps with two E-PG wedges. However, P-EN neurons and E-PG neurons that arborize in the same bridge glomerulus have shifted processes in the ellipsoid body (compare *Figure 1B and F*, *Wolff et al. [2015]*). Putative P-EN axons from the left side of the bridge arborize in tiles that are shifted clockwise with respect to the E-PG wedges while P-EN axons from the right side of the bridge are shifted counterclockwise (*Figure 1G*). We hypothesized that this anatomical shift, which bears a remarkable resemblance to network motifs proposed in ring attractor models explaining head direction system function (*Skaggs et al., 1995*; *Xie et al., 2002*; *Zhang, 1996*), could allow the P-EN population to move the E-PG bump in either direction depending on the fly's turns in the dark. Although the E-PG dendritic activity is localized in a single bump in the ellipsoid body (*Seelig and Jayaraman, 2015*), the projection patterns of the

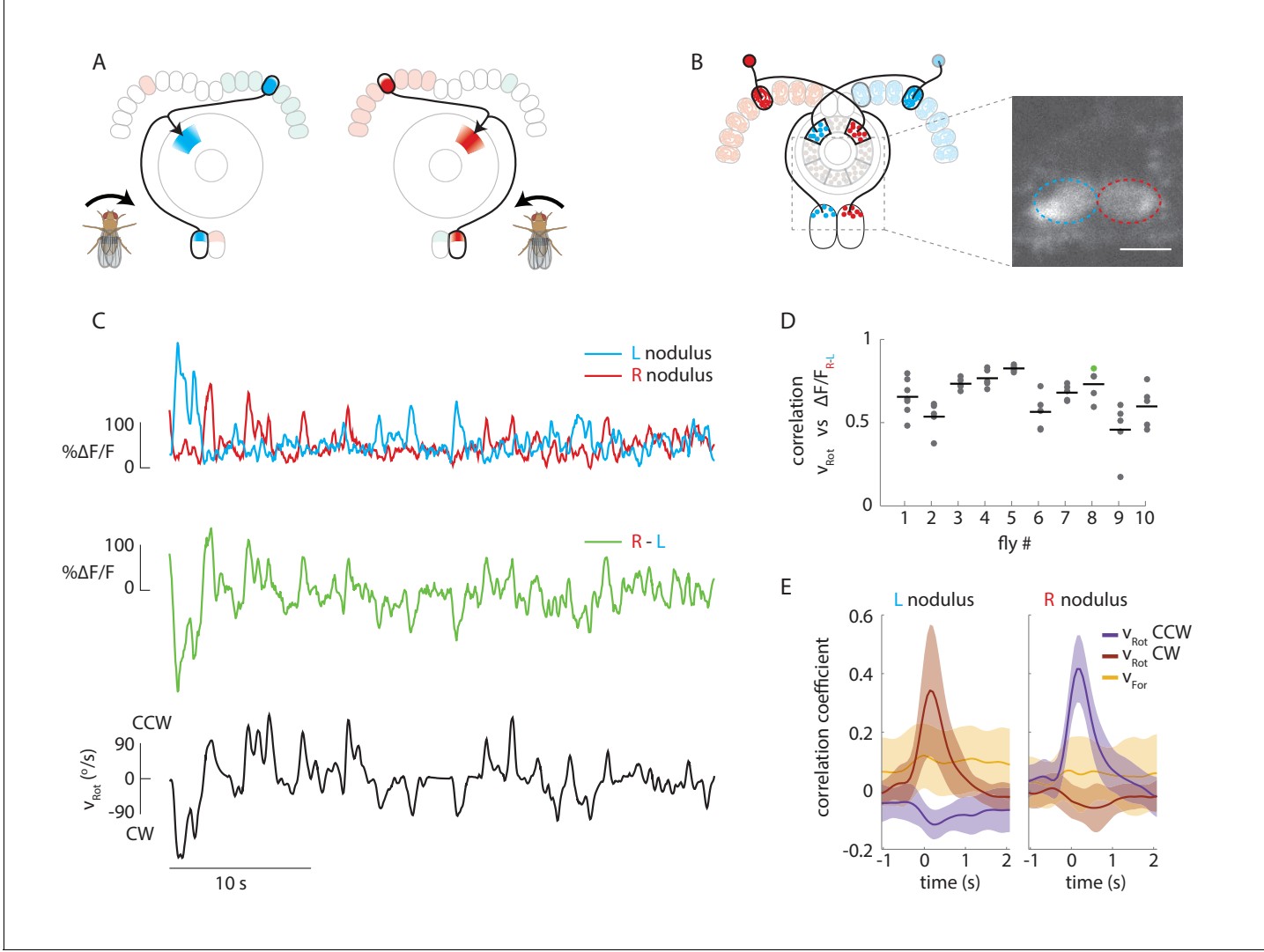

**Figure 2.** Calcium activity of P-EN neurons in the paired noduli correlates with fly's rotational velocity in darkness. (A) Schematic showing how a conceptual model (*Figure 1H*) to update heading representation would influence calcium activity in the paired noduli. Asymmetric activation of one or the other side of the bridge when the fly turns would, because of P-EN neurons' projection to the contralateral nodulus, result in activation of the opposite nodulus. (B) (Left) Schematic of the P-EN neurons, highlighting the left and right P-EN populations in red and light blue, respectively. Dashed rectangles mark the imaged region shown at right. (Right) Average of sample two-photon calcium imaging stack showing P-EN GCaMP6f signal in the left and right nodulus. ROIs used to calculate ΔF/F values are outlined by dotted ovals. For this and all other imaging experiments, F is defined as the lowest 10% of fluorescence levels during the trial. The scale bar is 20 μm. (C) P-EN calcium transients in the right (red) and left (blue) nodulus, and the difference between activity levels in the two noduli (green), compared to the fly's rotational velocity when walking in darkness (black). The velocity trace is convolved with the GCaMP6f time constant (see Materials and methods). (D) Correlation between the convolved rotational velocity and the difference between right and left noduli calcium activity across flies and across trials. Bars show mean values (N = 10 flies). The green point marks the example shown in C. (E) Linear regression analysis for calcium imaging in the noduli shows coefficients for the correlation of P-EN calcium activity in each nodulus with forward as well as clockwise and counter-clockwise rotational velocity.

E-PG neurons (*Figure 1B*) predict that this bump would manifest as two separate bumps in the protocerebral bridge, one on the right side and one on the left, each in synchrony with the bump in the ellipsoid body. These activity bumps in the bridge could then be passed to P-EN neurons whose dendrites co-localize with E-PG axons (*Figure 1H*, top). If P-EN activity on different sides of the protocerebral bridge were asymmetrically modulated by angular velocity input when the fly turned one way or the other, this, in turn, would cause P-EN axonal projections in the ellipsoid body to be asymmetrically activated on one or the other side of the existing E-PG bump (*Figure 1H*, middle). If the

P-ENs make excitatory connections with the E-PGs, this asymmetric activation would then pull the E-PG bump in one direction or the other. Thus, if the fly turned left (counterclockwise), and P-EN neurons on the left side of the bridge were more strongly excited, their projections in the ellipsoid body would then pull the E-PG bump clockwise, maintaining an appropriate representation of heading (*Figure 1H*, bottom, *Video 1*).

This conceptual model rests on three key assumptions. First, that P-EN activity encodes angular velocity, with mirror-symmetric tuning profiles for the left and right subpopulation. Such angular velocity coding has been reported in extracellular recordings of unidentified neurons in the cockroach central complex (*Guo and Ritzmann, 2013*; *Martin et al., 2015*). We tested this assumption by recording the population calcium activity of P-EN neurons in head-fixed flies walking on an air-supported ball (*Seelig et al., 2010*) as well as by recording the electrophysiological activity of individual P-ENs. Second, that activity in P-EN subpopulations on both sides of the protocerebral bridge localizes in a bump that conjunctively encodes both the fly's rotational velocity and its heading, which we assess through electrophysiological recordings and population calcium imaging. Third, that P-EN and E-PG neurons are functionally connected to one another, which we assessed anatomically, using immunohistochemical staining of presynaptic markers, and functionally, using a combination of optogenetics and two-photon calcium imaging in an ex-vivo preparation. This functional circuit architecture predicts a specific phase relationship between the two neural populations, with, for example, the P-EN population inheriting a bump of activity from the output of the E-PG population in the protocerebral bridge and the P-EN activity bump leading the E-PG bump in the ellipsoid body during turns. To examine this more fine-grained prediction, we recorded the population activity of P-EN and E-PG neurons simultaneously in the walking fly with two-color, two-photon calcium imaging. We found evidence in support of most of these assumptions, which enabled us to formalize our understanding in a firing rate model of this circuit mechanism that included a key additional component — recurrent inhibition. Finally, we validate a core assumption of our model by demonstrating that the population activity of E-PG neurons depends on synaptic input from P-ENs by conditionally blocking P-EN synaptic output using the temperature-sensitive dynamin mutation, shibire^TS (abbreviated shi^TS).

## Results

### Calcium activity of P-EN neurons in the noduli correlates with angular velocity

To test our first assumption, namely that P-EN neurons mirror-symmetrically encode angular velocity in the right versus left side of the protocerebral bridge, we expressed GCaMP6f in P-EN neurons (under the control of the R37F06-GAL4 driver) and imaged the calcium activity of the left and right P-EN subpopulations in the noduli, a third, paired structure that is innervated by these neurons (*Wolff et al., 2015*, *Figure 2A*). All P-EN neurons whose dendrites project to glomeruli on the right side of the bridge innervate the left nodulus, whereas those on the left side of the bridge project to the right nodulus. The proximity and compact nature of the noduli enables unambiguous assignment of activity to the left and right P-EN populations with a high signal-to-noise ratio (*Figure 2B,C*).

P-EN population calcium activity in the noduli of flies walking in the dark was strongly modulated by the fly's angular velocity. When the fly turned clockwise, P-EN activity increased in the

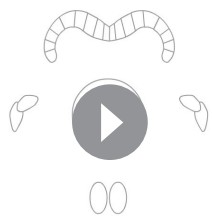

**Video 1.** Cartoon animation of E-PG and P-EN compass function during a turn. Activity of E-PG neurons is shown in red, and that of P-EN neurons in blue. Animation shows the interaction of the two populations during a turn, and highlights how heading-dependent E-PG input combines with turn-direction-dependent angular velocity input to trigger P-EN activity in the appropriate protocerebral bridge glomeruli. This activity, in turn, activates E-PG neurons in ellipsoid body sectors neighboring those that the bump originally inhabited, thereby updating the heading representation.

left nodulus, and vice versa. This led to an apparent flip-flopping of activity between the two noduli as the fly made turns in either direction (*Figure 2C*). Thus, the difference between calcium transients evoked in the two noduli was correlated to the fly's angular velocity (Pearson's R = 0.65 ± 0.14, mean ± SD, N = 10 flies, p<0.001 in all cases, *Figure 2C,D*, see Materials and methods). In fact, noduli calcium responses were dominated by the fly's angular velocity and largely indifferent to the fly's forward speed, as revealed by generalized linear regression analysis (*Figure 2E*, see Materials and methods). Thus, P-EN population activity is mirror-symmetrically tuned to angular velocity, with neurons innervating the right side of the bridge (and therefore the left nodulus) preferentially responding to clockwise rotation and vice versa, matching the requirements of our conceptual model (*Figures 1H* and *2A*).

## Two P-EN subpopulations mirror-symmetrically encode the fly's rotational velocity

To understand how individual P-EN neurons respond as the fly turns, we performed somatic loose patch and whole cell patch clamp recordings from identified neurons in tethered walking flies expressing GFP under control of the same GAL4 driver as used for calcium imaging, R37F06 (*Figure 3A*). We filled cells with Alexa dyes and confirmed neuron identity and soma location by epifluorescence imaging and/or post-hoc confocal imaging of PFA-fixated brains (*Figure 3A*, see Materials and methods). We let head-fixed flies walk in darkness during the recording, or, alternatively, linked the rotational component of their movements on the ball to the angular position of a 15° wide vertical stripe (visual closed loop, see Materials and methods).

As hinted at by P-EN population imaging in the noduli, the electrical activity of individual P-EN neurons was strongly modulated by rotational velocity. We found that P-ENs segregated into two anatomically defined subpopulations: Neurons with their soma in the right hemisphere increased their activity during turns to the right and decreased it during turns to the left, whereas neurons residing in the left hemisphere showed the inverse activity profile (*Figure 3B,C* and *Figure 4A,B*). Note that P-EN responses to changes in rotational velocity were similar in darkness and visual closed loop conditions (see *Figure 4—figure supplement 1A,D,E,H*), which allowed us to pool data for all population statistics presented in *Figure 4*. Consistent with our previous observations, P-ENs were generally not tuned to forward velocity (see Materials and methods and also *Figure 4—figure supplement 1B*). We next quantitatively characterized P-EN responses during the fly's turns by computing rotational velocity tuning curves. For each cell, we sorted rotational velocities of all walking periods into 12°/s bins and fit the mean P-EN spike rate across bins with a weighted sigmoidal function (*Figure 4A*; mean $R^2$ = 0.87 ± 0.01, N = 12, see also *Figure 4—figure supplement 1C*; see Materials and methods for details). On average, P-EN spike rates increased by 5.6 Hz during fast turns in the preferred compared to fast turns in the non-preferred direction, but there was considerable heterogeneity across cells ('rate modulation': 5.6 ± 3.7 Hz, N = 12; *Figure 4B* and *Figure 4—figure supplement 1D*). Likewise, P-ENs were heterogeneous in the range of rotational velocities over which their activity was modulated ('P-EN bandwidth': 145 ± 82°/s, N = 12). Overall, however, P-EN bandwidth corresponded well with the range of rotational velocities displayed by turning flies (*Figure 4—figure supplement 1E*). We also noted that the P-EN bandwidths of the left and right P-EN subpopulations were largely overlapping (*Figure 4C*, left), so that the inflexion points of their tuning curves (the rotational velocities for half-maximal P-EN activation) were clustered around 0°/s (*Figure 4C*, right). Thus, when the fly is walking, the total activity of the P-EN population is roughly constant, with the contributions of the left and right P-EN subpopulations depending on the fly's momentary rotational velocity (*Figure 4—figure supplement 1F*).

To examine whether increases in P-EN activity preceded upcoming turns, we computed spike-triggered averages (STAs) of instantaneous rotational velocity during periods of significant turning (with instantaneous rotational velocities greater than 40°/s, see Materials and methods). Across all P-ENs, we found that spikes occurred well after the peak of the rotational velocity, on average by 123 ± 43 ms (N = 12) (*Figure 4D*). In an alternative approach, we used generalized linear regression to fit the instantaneous P-EN spike rate with the past, present, or future rotational velocities by introducing a series of lags between the two parameters (*Figure 4—figure supplement 1G*, see Materials and methods). In all P-ENs, the maximum difference between left and right turn correlation coefficients occurred at positive temporal lags (130 ± 73 ms, N = 12), confirming that P-EN spiking activity changed as a response to turns instead of preceding them (*Figure 4—figure supplement*

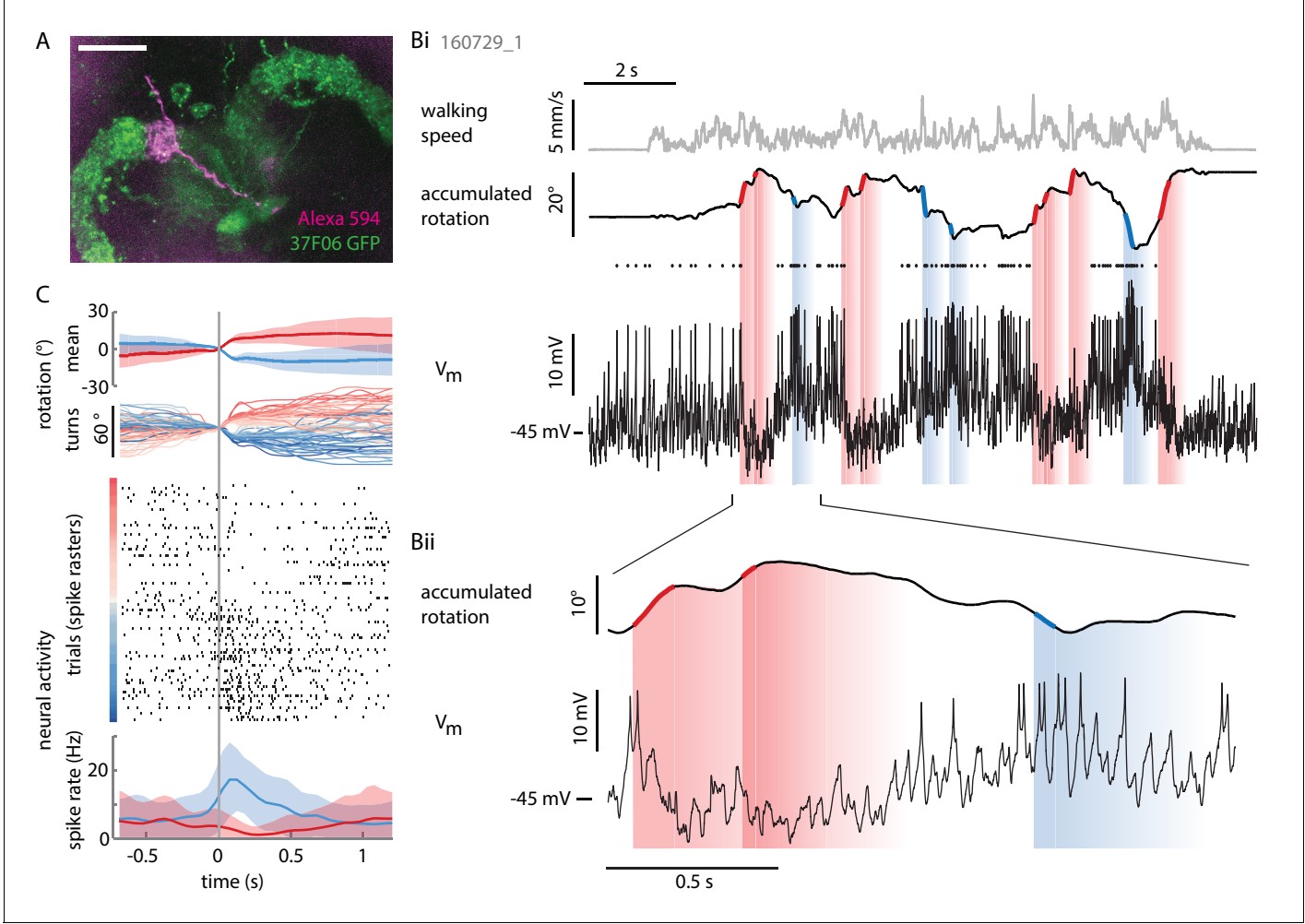

**Figure 3.** Individual P-ENs are tuned to the fly's rotations. (**A**) Z-projection of confocal image of a single P-EN neuron filled with Alexa594 (magenta) during intracellular whole-cell patch clamp recording in R37F06-GAL4 flies expressing GFP (green). Scale bar: 20 μm. (**B**) Example recording of a P-EN with soma location and protocerebral bridge innervation in the right hemisphere. The fly was walking in darkness. (**Bi**) Plotted are the fly's translational velocity (top row, gray) and accumulated rotation (second row, black), along with the cell's membrane potential (third row, black). Spike times are indicated with dots. Epochs of fast rotations are highlighted in red and blue on the rotation trace (left and right rotations, respectively). Since the change in P-EN activity often outlasted the duration of a turn, the vertical blue and red shaded regions indicate a time window of ~500 ms in which the neuron's activity reflected these prior fast rotations. (**Bii**) Expanded view, highlighting the P-EN neuron's hyperpolarization in response to rotations in the contralateral direction (for this neuron, left, red) and depolarization with spiking in response to rotations in the ipsilateral direction (for this neuron, right, blue). (**C**) All of the epochs during which the fly's angular velocity exceeded 80°/s were temporally aligned and color-coded by turn amplitude (right: red, left: blue). The top panel shows the mean rotation in each direction, the second panel the individual turns. Note that the time point '0' denotes when the angular velocity exceeds 80°/s, which is not the time of turn onset, since rotational velocities vary smoothly and continuously. The neural activity is plotted as a raster plot for individual turns (third panel) and as averaged spike rates (bottom panel).

*1H*). Overall, individual P-ENs encode the fly's recent rotational velocity, with left and right P-ENs segregating into two subpopulations with mirror-symmetric tuning profiles.

## Single-cell correlates of heading tuning in P-ENs

Since our conceptual model relies on the P-ENs to also be tuned to the fly's heading (*Figure 1H*), we next asked if the patch-clamp recordings would reveal such properties. To avoid complications caused by error accumulation in flies walking in the dark (*Seelig and Jayaraman, 2015*), which would result in a drift of the neuron's preferred heading angle over time, we first restricted this analysis to experiments with closed loop visual feedback (see Materials and methods). We constructed two-

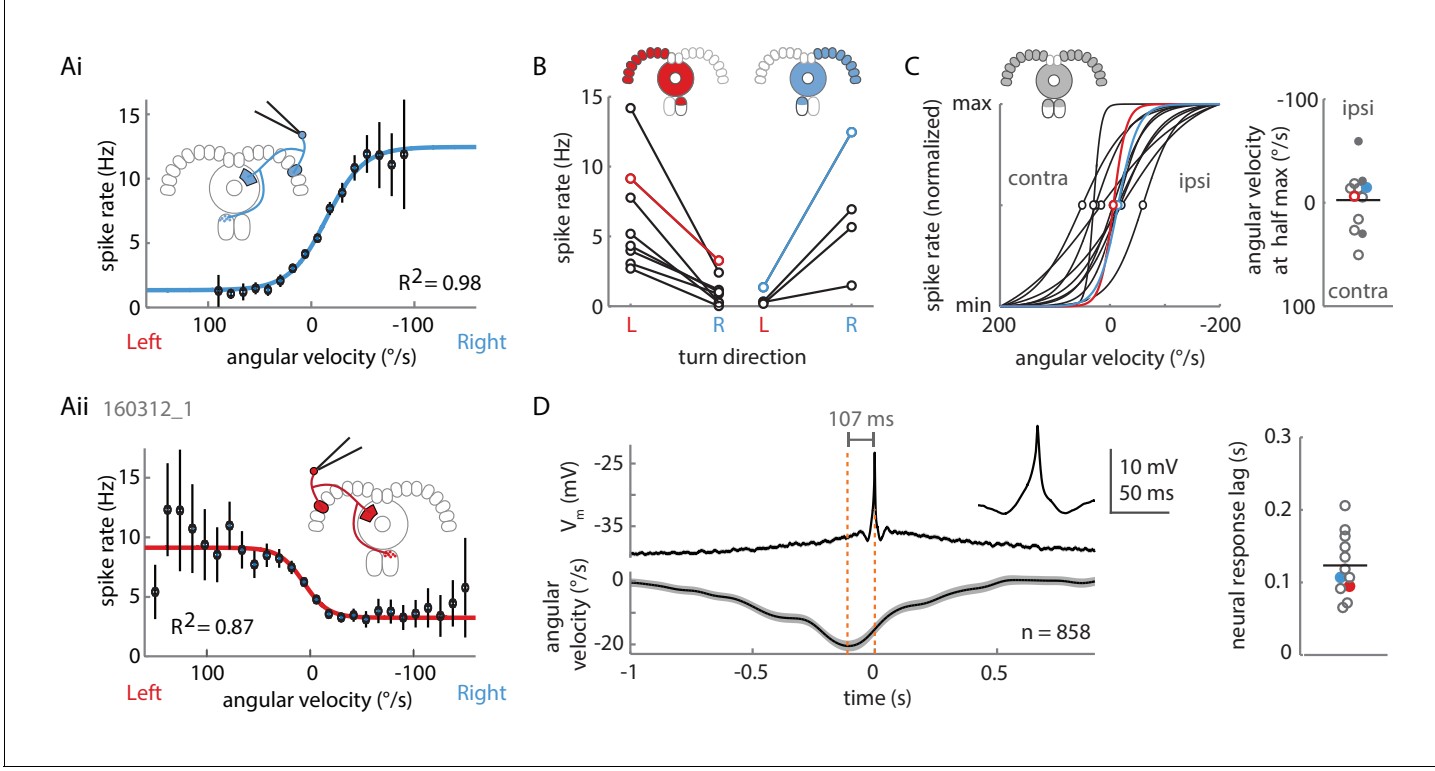

**Figure 4.** Two P-EN subpopulations mirror-symmetrically encode the fly's rotational velocity. (A) Example tuning curves of P-EN spike rate to the fly's rotations as the fly walks in darkness. Angular velocities were binned in 12°/s bins and a sigmoid was fitted with bin counts used as weights (see Materials and methods for details). Mean spike rate and 95% confidence interval in black, sigmoidal fits in blue and red. (Ai) Tuning curve for a right P-EN neuron (same as in *Figure 3*). P-EN membrane potential changes were similar to the observed changes in spike rates (see *Figure 4—figure supplement 1C* for $V_m$ tuning curve). (Aii) Tuning curve for a left P-EN neuron (see *Figure 4—figure supplement 2A* for example trace). (B) Fitted spike rates for the flies' turns to the left versus right illustrate mirror-symmetric tuning properties of the left and right P-EN subpopulations. Spike rates were computed either at saturation or at an absolute rotational velocity of 200°/s, whichever was lower. Example cells from panel A are color coded. (C) Encoding of rotations by the two subpopulations is mirror-symmetric yet overlapping, that is, each P-EN subpopulation encodes rotations in both directions. Left: Normalized tuning curve fits for all P-EN neurons. Left hemisphere P-EN curves have been reflected for simplicity. Neurons plotted in A are color coded for comparison. Open circles mark each sigmoid's half-maximum (inflexion point). Right: On average, the P-EN neurons' receptive field for rotations is centered around 0°/s, where P-EN's are half-maximally activated. Closed and open circles in the right subplot represent inflexion points of right and left P-EN tuning curves, respectively. Example cells from panel A are in blue and red. (D) Spike-triggered averages of angular velocities were constructed to characterize P-EN timing. Since P-EN neurons tend to spike at rest, all spikes with rotational velocities not exceeding 40°/s at any time in a one second window around the spike were excluded. Left: Membrane potential is plotted at top, angular velocity at bottom. Inset shows a magnification of the average spike shape. Right: The peak of the angular velocity precedes the spike in all P-ENs recorded. Example cells from panel A are in blue and red.

The following figure supplements are available for figure 4:

**Figure supplement 1.** Extended data on P-EN rotational velocity tuning.

**Figure supplement 2.** Loose patch recordings.

dimensional tuning maps of P-EN spike rates to rotational velocity and virtual heading, which revealed localized peaks in firing rate (*Figure 5A*). We collapsed these maps along either of the two axes to fit the P-EN's tuning to the fly's virtual heading with a compound von Mises function and its tuning to rotational velocity with a sigmoidal function, as before (*Figure 5A*, see Materials and methods). For each cell, we also calculated indices for rotation and heading tuning (*Figure 5B*, see Materials and methods), which were, in all but one case (p=0.0676), larger than the 95th percentile of the shuffled control distribution.

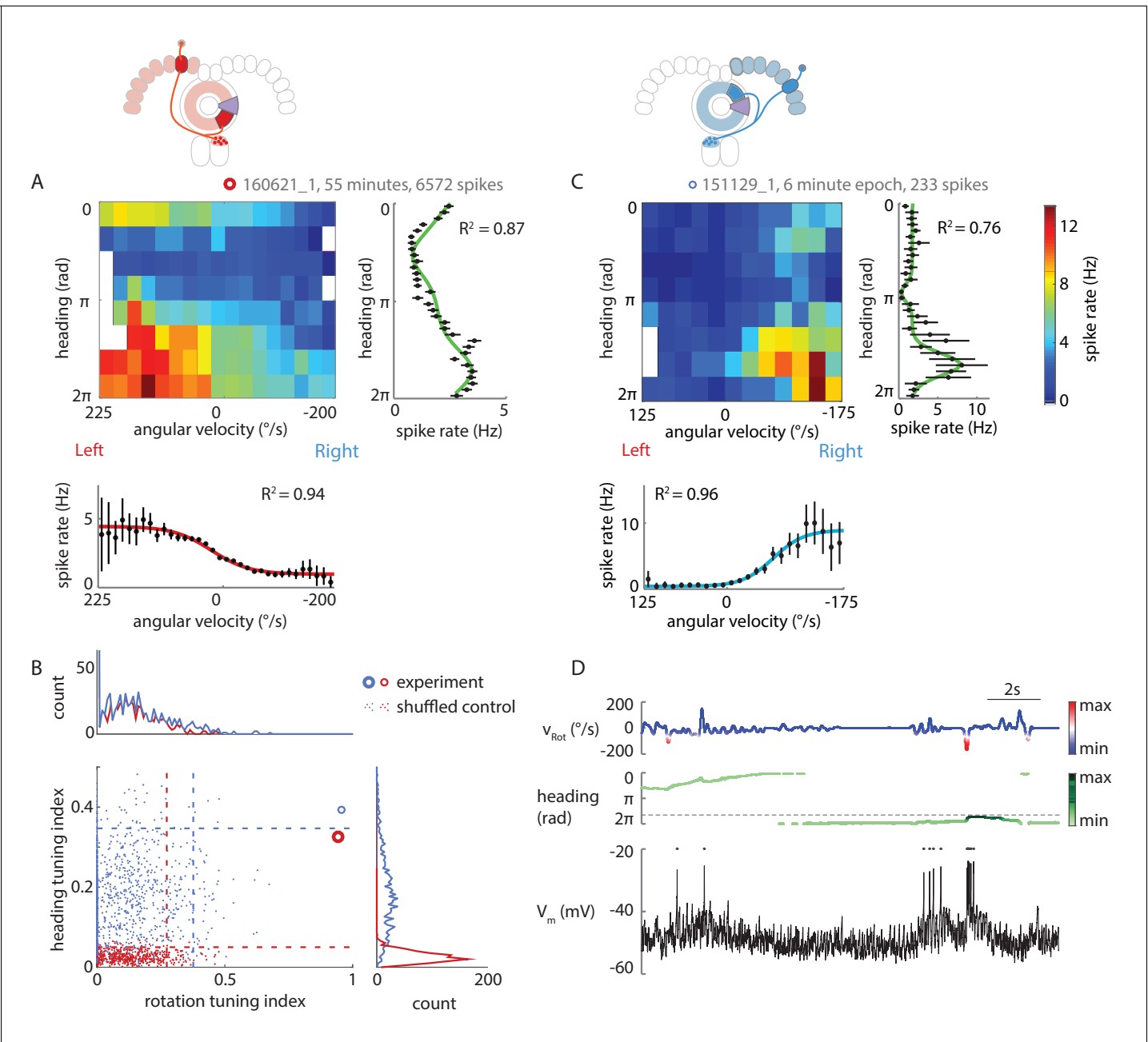

**Figure 5.** P-ENs are conjunctively tuned to angular velocity and heading. (**A**) Two-dimensional tuning properties of a P-EN's spike output averaged over the entire duration of the experiment. The cell was located in the left hemisphere (schematic at top). The fly walked with closed loop visual feedback. The heat map (upper left) shows spike rates as a function of angular velocity and virtual heading. A tuning curve was fitted to the fly's virtual heading angle (upper right) as well as the fly's angular velocity (lower left) (see Materials and methods). Black: mean rate and 95% confidence intervals. Tuning curves are least squares fit to the sum of two von Mises functions for heading (green) and to a sigmoidal function for rotations (red). (**B**) Quantification of the heading tuning index (mean vector length of spike rates across virtual heading angles) and the rotation tuning index ($R^2$ values for sigmoidal tuning curve fit) for the two example cells plotted in **A** and **C** (bold red open circle and blue open circle, respectively). Shuffled controls (see Materials and methods) are plotted as dots and dotted lines mark 95th percentile boundaries (plotted in red and blue for the two example cells). Histograms of shuffled data indices are plotted at top and at right following the same color code. Tuning indices for all cells are shown in *Figure 5—figure supplement 1C*. (**C**) Two-dimensional tuning properties of a P-EN's spike output during a brief epoch that still sufficiently sampled the two-dimensional parameter space. The cell was located in the right hemisphere (schematic at top). The fly walked with a closed loop visual feedback. The heat map (upper left) shows spike rates as a function of angular velocity and virtual heading. Colorbar at right. Tuning curves to the fly's virtual heading angle and angular velocity are plotted analogous to **A**. (**D**) Example traces for the epoch plotted in **C** illustrating tuning to rotational velocity as well as virtual heading. The two parameters are color coded according to the neural activity expected from the tuning curves plotted in panel **C**.

*Figure 5 continued on next page*

*Figure 5 continued*

The following figure supplement is available for figure 5:

**Figure supplement 1.** Extended data on P-EN conjunctive tuning.

P-EN heading tuning was generally stable over time, although we occasionally observed drifts in the neuron's preferred heading angle (*Figure 5—figure supplement 1A*). We assessed tuning stability more quantitatively by analyzing short epochs (5–10 min) that sampled a sufficient range of rotational velocities ($\geq 250°/s$) across all virtual heading angles (see Materials and methods). We identified seven such epochs in six different cells recorded in flies walking with closed loop visual feedback as well as in darkness (see *Figure 5C,D* for an example and *Figure 5—figure supplement 1C* for tuning indices, which were, for all but one epoch, larger than the 95[th] percentile of the shuffled control distributions). For all epochs recorded in flies walking with visual feedback, we found that the P-ENs' preferred heading angles during the epochs differed by only $6 \pm 5\%$ from the average of the entire experiment (population mean, N = 4). As a second line of evidence, we found the P-EN bump width to be roughly similar for short epochs and across whole experiments, suggesting that a neuron's preferred heading angle did not drift much over time (average tuning curve width at half-maximal P-EN heading modulation: $56 \pm 20°$ (N = 7 epochs) versus $76 \pm 35°$ [N = 5 experiment averages]). Overall, we found P-EN neurons to be tuned to the fly's heading as well as the fly's rotations, both when flies walked under visual closed loop conditions and in darkness (*Figure 5—figure supplement 1C*).

## Conjunctive tuning to heading and rotation is sharpened within individual P-ENs

Having established that P-EN neurons are conjunctively tuned to the fly's heading and rotations, we examined our recordings for hints of integrative computations within individual neurons. Specifically, we began by asking whether P-EN activity was tuned to one of the two parameters even in the absence of contribution from the other. *Figure 6A–B* depicts an example of strong P-EN tuning to rotational velocity as well as to virtual heading for a fly walking in darkness. To quantitatively assess whether tuning to one parameter depended on the other, we computed the rotation and heading tuning indices separately for the respective preferred and non-preferred conditions, i.e. one rotation index each for the fly's turns within the preferred versus non-preferred heading quadrant and one heading index each for when the fly rotated in its preferred versus non-preferred turn direction. These tuning indices were not significantly different (*Figure 6C*, left). Along the same lines, the preferred heading angle of individual P-ENs (the mean vector angle of spike rates across headings) was similar for preferred and non-preferred rotations (*Figure 6C*, bottom right). Following the same logic, we also fit conditional tuning curves to subsets of the parameter space (see Materials and methods for details). As evident from the example (*Figure 6D*) and the population data (*Figure 6E*), strong conjunctive tuning was apparent in the pronounced differences in rate modulation between preferred and non-preferred conditions. Nevertheless, we found P-EN spiking to be tuned to rotational velocity in non-preferred heading quadrants and to heading during non-preferred rotations (*Figure 6E* left; rotation tuning at non-preferred heading angles: $2.3 \pm 2.1$ Hz, heading tuning during non-preferred rotations: $1.2 \pm 1.1$ Hz; one sample t-test: p=0.026 and p=0.043, respectively; N = 6 for both comparisons). The pronounced conjunctive tuning we observed at the level of spike rates was less apparent in the subthreshold activity of P-EN neurons (*Figure 6—figure supplement 1*). In contrast to the spike rates, for which rate modulation in response to one parameter strongly depended on the other, quantitative modulation of the membrane potential did not show significant mutual dependence between the two parameters (*Figure 6—figure supplement 1C*, see Materials and methods for details). These observations suggest not only that tuning of P-EN output is strongly conjunctive, but also that this property may be sharpened within single P-ENs.

## A recurrent loop between P-ENs and E-PGs

We next sought to test the third pillar of our conceptual model, that the E-PG and P-EN neurons are connected in an excitatory loop. Light level analysis of P-EN neuron morphology suggests that they

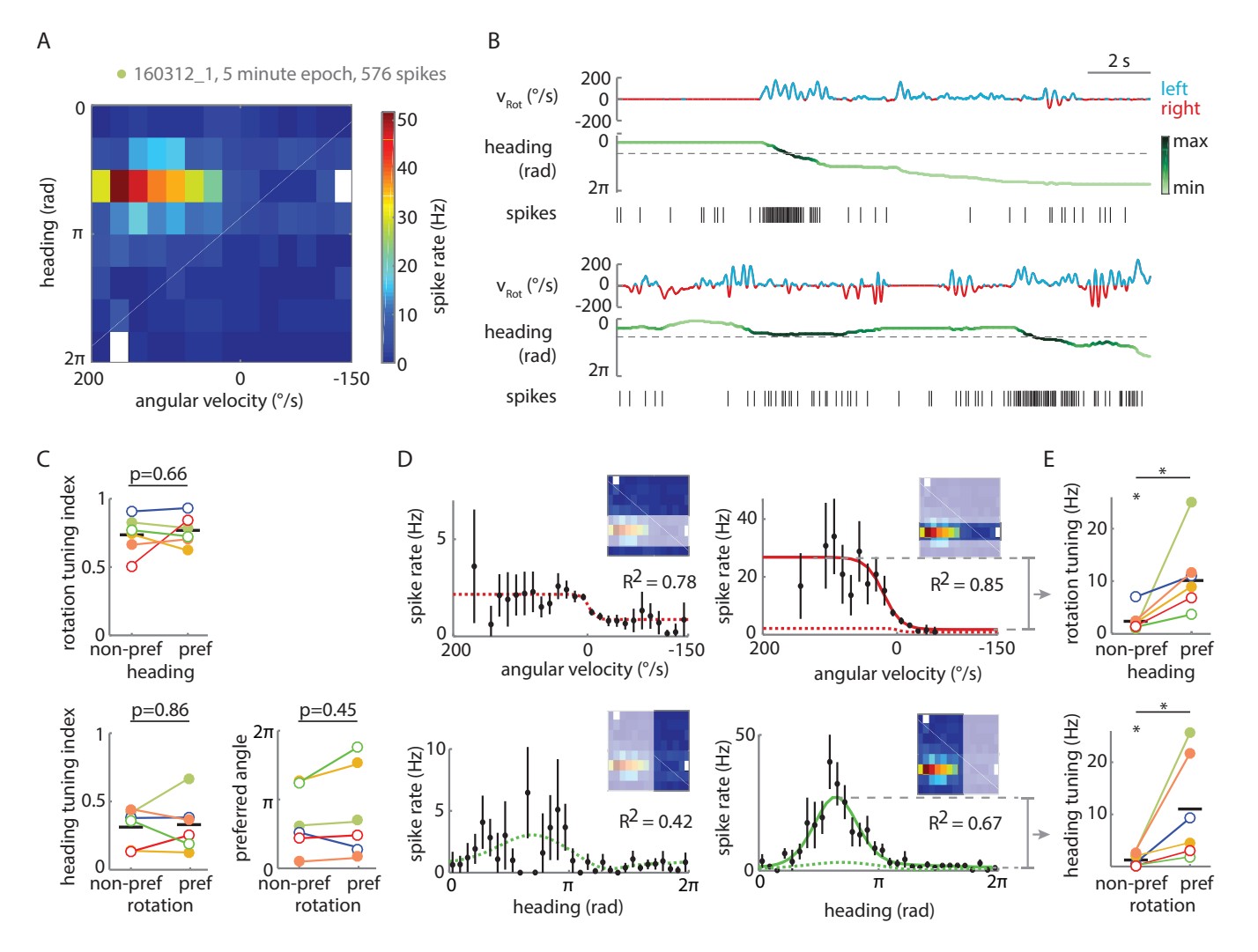

**Figure 6.** Tuning of P-EN spiking activity is evident even for non-preferred heading and rotational velocities. (**A**) Heat map shows conjunctive tuning of a P-EN's spike output to angular velocity and virtual heading. The fly walked in darkness. (The recording is the same as shown in *Figure 4Aii* and *Figure 4—figure supplement 2*). (**B**) Example traces from that recording illustrating conjunctive tuning to rotational velocity and virtual heading. The fly's rotational velocity is color-coded by preferred (left, blue) and non-preferred (right, red) turn direction. The fly's heading is color coded according to the neural activity expected from the heading tuning curve computed from A (see Materials and methods, light green: low, black: high). (**C**) P-EN spike output is tuned to rotations irrespective of the fly's heading (top, $R^2$ value of sigmoidal tuning curve fit) and informative about the fly's heading irrespective of rotations, as evident both from the heading tuning index (bottom left, mean vector length of spike rates across virtual heading angles) and the relative stability of the preferred heading angle (mean vector angle). P-values are results of paired two-sample t-tests. (**D**) Conditional tuning curves for the cell plotted in **A**. Schematics illustrate the subset of data used to construct the tuning curves. Mean and 95% confidence intervals are plotted in black, fits for preferred conditions in solid and for non-preferred conditions in dotted lines. Top: Tuning to angular velocity given heading (left: rotations in non-preferred heading quadrant, right: rotations in preferred heading quadrant). Bottom: Tuning to heading given turn direction (left: non-preferred turn direction, right: preferred turn direction). (**E**) Strong P-EN conjunctive tuning is apparent from the increased rate modulation for tuning to one parameter in the preferred range of the other. Top: Difference in fitted spike rates at −150 and 150°/s for each condition (two sample t-test: p=0.049, N = 6). Bottom: Difference in fitted spike rates at the heading angles corresponding to the peak and the trough of the unconditional heading tuning curve (two sample t-test: p=0.042, N = 6). Tuning to rotations in non-preferred heading quadrants (top left) and tuning to heading during non-preferred rotations (bottom left) is, however, significantly different from zero (one sample t-test: p=0.026 and p=0.043, respectively). Filled circles are recordings in darkness; open circles, in visual closed loop. Color code is the same as used in *Figure 5—figure supplement 1C*. p<0.05.

The following figure supplement is available for figure 6:

**Figure supplement 1.** P-EN spiking activity shows stronger conjunctive tuning than membrane potential.

likely have post-synaptic specializations in the bridge and pre-synaptic boutons in the ellipsoid body, while E-PG morphology indicates the opposite polarity (*Wolff et al., 2015*), making it possible for the two cell types to be mutually connected. We sought histochemical confirmation of this polarity by expressing the presynaptic reporter synaptotagmin-smGFP-HA (*Aso et al., 2014*) in GAL4 lines that drive expression in P-EN and E-PG neurons, respectively. Light level analysis of brains in which HA-tagged synaptotagmin was expressed in P-EN neurons revealed stronger labeling in the ellipsoid body and noduli than in the bridge (*Figure 7A*, 11/14 brains). In contrast, analysis of the E-PG targeted brains revealed high levels of HA-tagged synaptotagmin in the protocerebral bridge, but also in the ellipsoid body (at higher levels than in the ellipsoid body in 5/15 brains, at comparable levels in both neuropils in 10/15 brains, see *Figure 7B*). These results are consistent with the anatomical finding that the P-EN neurons receive input in the bridge and send output to the ellipsoid body.

To confirm these key connections, we then performed functional connectivity experiments between the two neuronal subtypes in ex vivo preparations. We expressed the red-shifted channelrhodopsin CsChrimson (*Klapoetke et al., 2014*) in either the E-PG or the P-EN populations while expressing GCaMP6m or GCaMP6f in the putative downstream partner cell type. Optogenetic activation of P-EN neurons reliably evoked positive calcium transients in E-PG neurons (*Figure 7C*, see Materials and methods), suggesting that angular velocity information is indeed relayed to E-PGs via this pathway. However, activation of the E-PG population induced more variable responses in P-ENs. Although we observed responses consistent with excitatory connections when we used strong light intensities for optogenetic activation (*Figure 7D*), the same pairing evoked activation, inhibition, and a lack of response at different times with weaker light intensities (dotted line in *Figure 7D*, inset). Thus, the excitatory loop between E-PG and P-EN neurons may be both direct and indirect, and the connection from the E-PG neurons onto the P-EN neurons likely also recruits inhibition. Overall, we hypothesize that other neurons in the ellipsoid body and protocerebral bridge (*Wolff et al., 2015*) likely influence the exchange of information between these two neuron types.

## P-EN calcium activity in the protocerebral bridge follows E-PG activity

Consistent with our conceptual model (*Figure 1H*), our electrophysiological results showed that individual P-EN neurons are tuned to heading and rotational velocity. Further, our functional connectivity results are supportive of a recurrent loop between the P-EN and E-PG populations, albeit perhaps with greater complexity than proposed by our simple conceptual model. This loop would, in the model, imply that the P-EN population inherits a bump of activity from the output of the E-PG population in the bridge, and that P-EN activity leads the E-PG bump in the ellipsoid body. To understand the spatiotemporal relationship between the two populations, we performed two-color calcium imaging (*Dana et al., 2016*) in flies walking in darkness. We expressed GCaMP6f in one population and jRGECO1a, a red indicator, in the other (*Figure 8A*). We tested for bleed-through and cross talk between the two channels (*Sun et al., 2017*), and verified clean color channel separation (see *Figure 8—figure supplement 1* and Materials and methods for details; *Sun et al., 2017*). We also checked that our results were qualitatively consistent across the calcium indicator combinations we used and pooled the data for both color combinations in the results presented below, color-coding data according to the calcium indicator with which they were obtained in *Figure 8*.

Imaging in the protocerebral bridge revealed two bumps of activity in the E-PG and P-EN neurons, one on each side of the bridge (*Figure 8B*), as predicted by our working model. To analyze the imaging stacks, we subdivided each half of the protocerebral bridge into nine regions of interest, one for each glomerulus (*Figure 8C*) and compared the activity in those regions over time to the rotational velocity for single trials (*Figure 8D*) and across trials (*Figure 8E*). For both neuron types, the bump intensities, but not widths, increased with increasing angular velocity across trials (*Figure 8E,F*). Although the P-EN bump intensity increased for both turn directions, which was unexpected, the increase was larger for ipsilateral turns than for contralateral turns (*Figure 8—figure supplement 2A*, top), a mirror-symmetry consistent with imaging results in the noduli (*Figure 2*) and with single cell electrophysiology (*Figure 4B*). The bump half-widths spanned ~2 glomeruli, which, if projected to the ellipsoid body, would lead to an activity width of 90 degrees. Consistent with our model, in which E-PG neurons carry the heading representation to the protocerebral bridge (*Figure 1H*), and in contrast with the activity of the P-EN subpopulations, there was no difference between E-PG activity for ipsilateral versus contralateral turns at any turn velocity (*Figure 8—figure supplement 2A*, bottom).

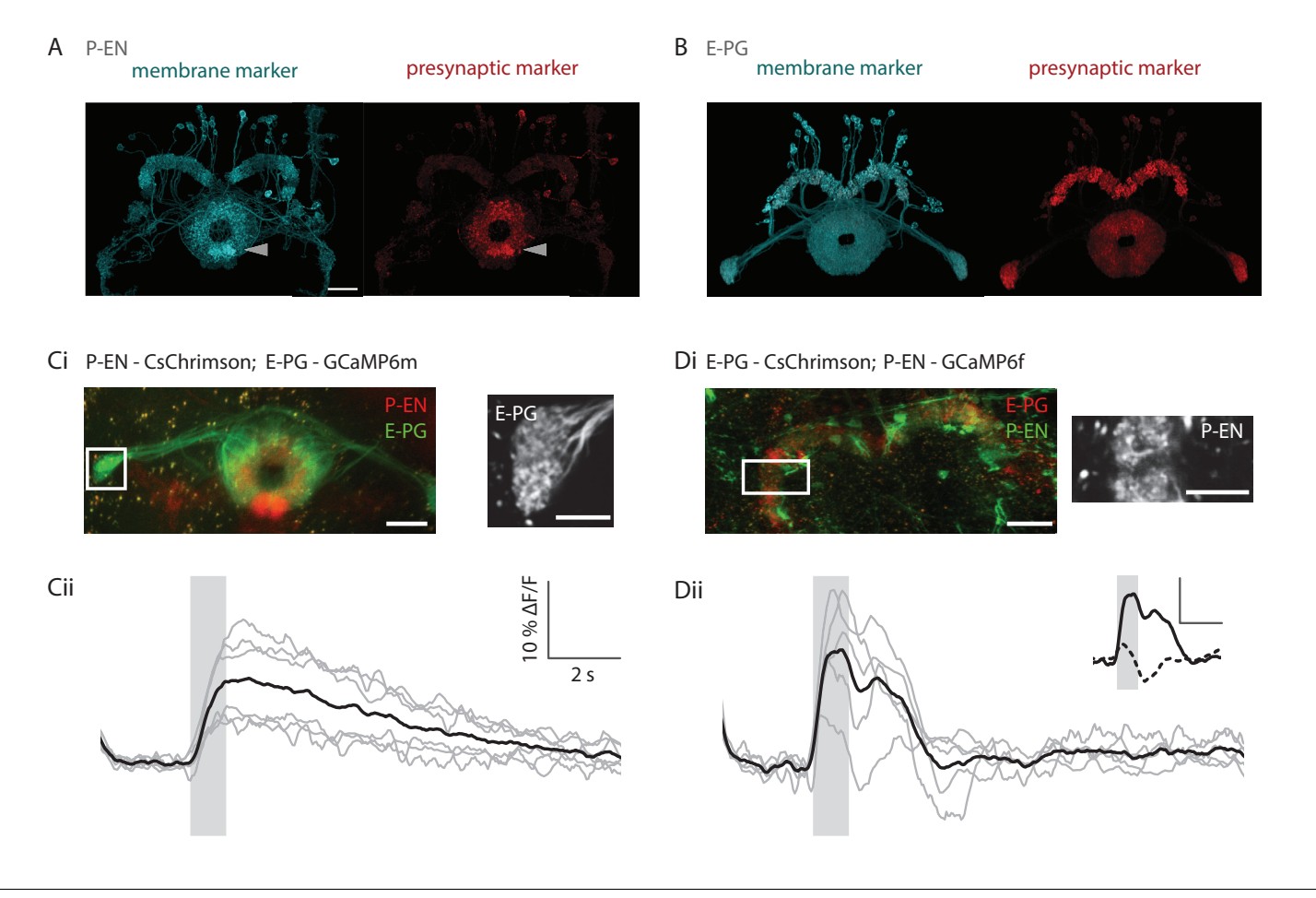

**Figure 7.** Functional connectivity of E-PG and P-EN neurons. (**A**) P-EN neuron membranes and presynaptic sites are labeled by expressing myr-sm::GFP and synaptotagmin in VT008135-GAL4, a second genetic line that drives expression in the P-EN neurons. While additional central complex and central brain cell types are targeted in this line, only the P-EN neuron arborizes in the protocerebral bridge (see Materials and methods). Expression is evident in the protocerebral bridge, the ellipsoid body and the paired noduli (behind the ellipsoid body, labelled by arrowhead). Intensity of presynaptic labeling was strongest in the ellipsoid body. The scale bar is 20 μm and applies to A and B. Though the image stack was rotated in three dimensions to obtain the included view, the scale bar was obtained from a single plane of the imaging stack. (**B**) Analogous to **A**, but using R60D05-GAL4 to drive expression in the E-PG neurons. Although presynaptic labeling was most intense in the protocerebral bridge and gall, synaptotagmin was also detected in the ellipsoid body, suggesting E-PG presynaptic sites in that neuropil as well. (**C**) Functional connectivity experiment with CsChrimson-mCherry expressed in P-EN, and GCaMP6m in E-PG neurons. (**Ci**) (Left) Average projection of fluorescence from a dissected brain. Both E-PG and P-EN arborizations can be seen in the ellipsoid body. E-PGs furthermore project to the gall (box, left) and P-ENs to the noduli (paired structure at bottom). (Right) Average frame from imaging the E-PGs in the gall while activating the P-ENs. (**Cii**) Average response (in black) from six flies to a train of 2 ms, 590 nm light pulses delivered at 30 Hz and 50 μW/mm$^2$. Stimulation time indicated by gray area. Gray traces correspond to the average of 4 trials for individual flies. (**D**) Functional connectivity experiment with CsChrimson-mCherry expressed in E-PG and GCaMP6f in P-EN neurons. (**Di**) (Left) Average projection of fluorescence from a dissected brain. (Right) Average frame from imaging the P-ENs in the protocerebral bridge while activating the E-PGs. (**Dii**) Analogous to **Cii**, except that stimulation light intensity was increased to 500 μW/mm$^2$. (inset) Comparison of low stimulation intensity responses (50 μW/mm$^2$, dotted line) and the high stimulation responses shown in the plot below (solid line). These responses were acquired in different regions (low stimulation: noduli, high stimulation: protocerebral bridge), but results were consistent across neuropiles. The scale bars are scaled versions of those shown in **Cii**. All scale bars in **Ci**, **Di**: 10 μm.

To determine the spatiotemporal relationship of the E-PG and P-EN bumps, we carefully examined their relative positions over time and across rotational velocities. To facilitate this comparison, we simultaneously registered both populations with respect to the E-PG bump in the right half of the bridge. We then binned these registered traces by rotational velocity. Across velocities, the registered E-PG and P-EN activity bumps overlapped, but with the P-EN bumps slightly lagging the

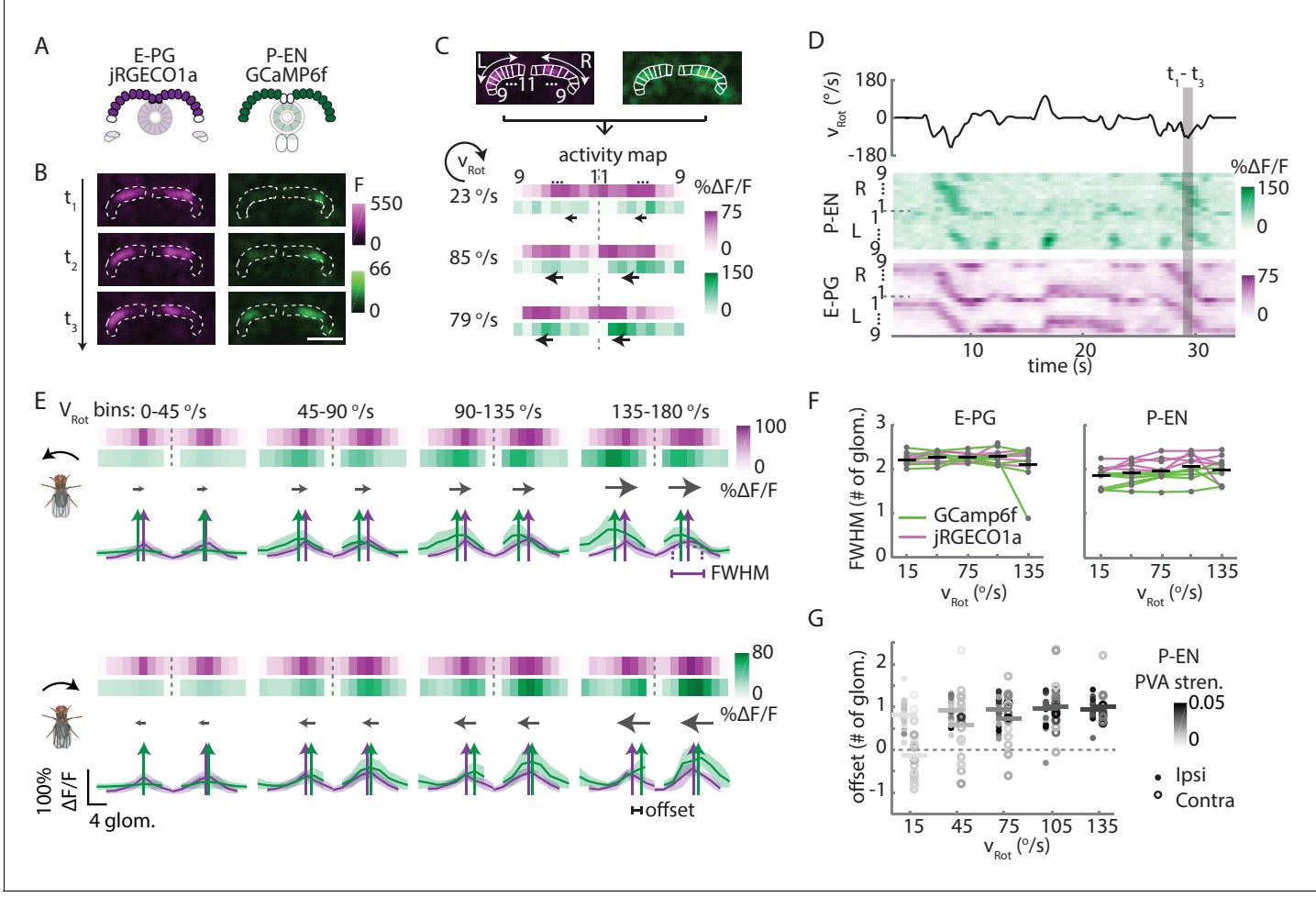

**Figure 8.** In the protocerebral bridge, P-EN calcium activity overlaps E-PG calcium activity with an offset. (**A**) Overview of a typical protocerebral bridge imaging experiment. jRGECO1a (shown in magenta), a red calcium indicator, is expressed in the E-PG neurons. GCaMP6f (shown in green), a green calcium indicator, is expressed in the P-EN neurons. Both populations are imaged simultaneously. (**B**) Maximum intensity projections of fluorescence from simultaneous imaging of E-PG and P-EN neural populations. Time points are 0.4 s apart. Scale bar: 50 μm. (**C**) ROI definition and ΔF/F projections for E-PG neurons (magenta) and P-EN neurons (green). The three time points shown in **B** are displayed at the bottom. The dotted line divides the right and the left half of the protocerebral bridge. Arrows indicate the direction of bump movement. (**D**) Protocerebral bridge activity over time for one trial for the E-PG and P-EN neurons. The fly's rotational velocity during the trial is shown in the black trace at top. The right and left side activity is concatenated. (**E**) Fluorescence transients binned by angular velocity and aligned to the peak of E-PG (magenta) activity in the right protocerebral bridge for five trials for one fly. The bar projections show the mean values while the cross-sectional plots show the mean and standard deviation. Colored vertical arrows show the PVA values of the traces. (**F**) Bump half widths (full width at half maximum, or FWHM) for P-EN neurons and E-PG neurons across flies and across velocity bins. The center velocity of each bin is shown, and the bin width is 30°/s. Points for individual flies are connected with colored lines indicating the calcium indicator tag. Normalized P-EN (E–PG) widths, obtained by dividing each width by the bump width for the given fly at 0–30°/s, were flat across velocities, with slopes of $-0.71 \pm 2.2 \times 10^{-3}$ ($0.98 \pm 0.68 \times 10^{-3}$) s/° for the green indicator and $0.36 \pm 0.68 \times 10^{-3}$ ($0.55 \pm 1.1 \times 10^{-3}$) s/° for the red indicator. (**G**) Difference in the population vector average (PVA) for the mean E-PG and P-EN calcium activity in the side of the protocerebral bridge ipsilateral and contralateral to a turn, plotted across flies and across angular velocity bins (N = 5 flies with the P-EN neurons tagged with GCaMP6f and E-PG neurons tagged with jRGECO1a, and N = 6 flies with the opposite calcium indicator combination). Each point represents the mean across trials for one fly and bars show the mean across flies. Individual dots are shaded to match P-EN PVA strength. At 120–150°/s, the offset is $0.96 \pm 0.40$ glomeruli for the ipsilateral side and $1.0 \pm 0.56$ glomeruli for the contralateral side. If the offset is instead calculated by finding the average PVA offsets of all of the individual traces (instead of the one PVA offset of the average trace), those values become $0.97 \pm 0.43$ and $0.46 \pm 1.0$ for the ipsilateral and contralateral sides, respectively. If the peaks are used instead of the PVA, the offsets at 120–150°/s are found to be $0.59 \pm 1.2$ and $0.36 \pm 1.3$ for the mean traces and $0.95 \pm 0.42$ and $0.41 \pm 0.91$ for the mean of the peak differences of the individual traces.

The following figure supplements are available for figure 8:

**Figure supplement 1.** The red and green channels are well separated.

*Figure 8 continued on next page*

*Figure 8 continued*

**Figure supplement 2.** Additional quantification of protocerebral bridge activity.

E-PG bumps (*Figure 8E,G*, see Materials and methods for further details on the registration and binning procedures). The lag led to bump offsets of approximately one glomerulus (the equivalent of 45 degrees in the ellipsoid body) at 120–150°/s, the fastest rotational velocity bin considered (and the bin with the greatest offset), whether measured by the population vector average or by peak difference (*Figure 8G*). This offset was also consistent across different fly lines (*Figure 8—figure supplement 2B*). While the substantial overlap between the two bumps suggests that the heading representation in the two populations is, in fact, roughly coincident, the observation that P-EN activity follows E-PG activity by one glomerulus on both sides of the bridge was not predicted by our simple model. This lag may be due to the subtleties of connectivity between the E-PG neurons and P-EN neurons (as hinted at, for example, by E-PG presynaptic specializations in the ellipsoid body [*Figure 7B*] and by the complex responses seen in our functional connectivity experiments [*Figure 7D*, inset]) or, simply, due to delays caused by neural time constants (see model below).

## P-EN calcium activity in the ellipsoid body leads E-PG activity

Next, we sought to investigate the relationship of the P-EN and E-PG population activity in the ellipsoid body, where, the conceptual model would suggest (*Figure 1H*), P-EN activity should lead E-PG activity. To do so, we once again performed two-color imaging of the two populations (*Figure 9A*). As in the protocerebral bridge, we observed bumps of activity in both neural populations (*Figure 9B*). We then subdivided the activity into 16 regions of interest, corresponding to the number of distinct E-PG arborization sectors in the ellipsoid body (*Figure 9B*, bottom; [*Wolff et al., 2015*]) and compared the activity in those regions over time (*Figure 9C*). Bump amplitudes, but not half-widths, varied with angular velocity (*Figure 9D,E*), and both bumps were ~100° wide (full width at half maximum, *Figure 9E*).

To compare E-PG and P-EN bump positions in the ellipsoid body across time, we performed a registration and binning procedure similar to the one used in the protocerebral bridge (see Materials and methods). Here, we used the E-PG bump in the ellipsoid body to align averaged calcium transients binned by the fly's angular velocity (*Figure 9F*, standard deviations shown in *Figure 9—figure supplement 1*). As the fly turned, the relative position of the E-PG and P-EN bumps around the ellipsoid body changed. At angular velocities less than 30°/s, all flies showed bump offsets of less than 6° (2.5 ± 2.1°, N = 10 flies, *Figure 9F,G*). However, at rotational velocities above 30°/s, P-EN population activity separated from E-PG activity, with the P-EN bump leading the E-PG bump around the ellipse (*Figure 9G*, *Video 2*). When the red indicator was expressed in the E-PG neurons and the green indicator in the P-EN neurons, the bump offset exceeded 15 degrees of separation for turns faster than 90°/s regardless of turn direction and regardless of whether the fly was angularly accelerating or decelerating (20.7 ± 11.7° at 150–180°/s). When, instead, the red indicator was expressed in the P-ENs, the lead-lag behavior was qualitatively similar, though the shift only rose to 7.3 ± 7.2°/s at 150–180°/s (*Figure 9—figure supplement 2*). As jRGECO1a's kinetics are different from those of GCaMP6f, a difference in the observed shift across color combinations is to be expected (*Dana et al., 2016*). Thus, we concluded that the P-EN bump leads the E-PG bump in the ellipsoid body during turns, consistent with the conceptual model.

## Firing rate model captures interaction of angular velocity and compass signals

Our conceptual model (*Figure 1H*) suggested that the anatomical offset between the E-PG and P-EN populations would move the activity bump in both populations during turns, thereby integrating rotational velocity to compute heading. However, it offered few predictions for how the feedback loops in the circuit might shape bump dynamics. In our two-color imaging experiments, we observed the hypothesized P-EN to E-PG activity offset, but the shift was small, between 10° and 30° at the highest angular velocities. Further, the E-PG activity in the bridge was consistently advanced from the P-EN activity by an offset that could exceed one glomerulus, which is half of the

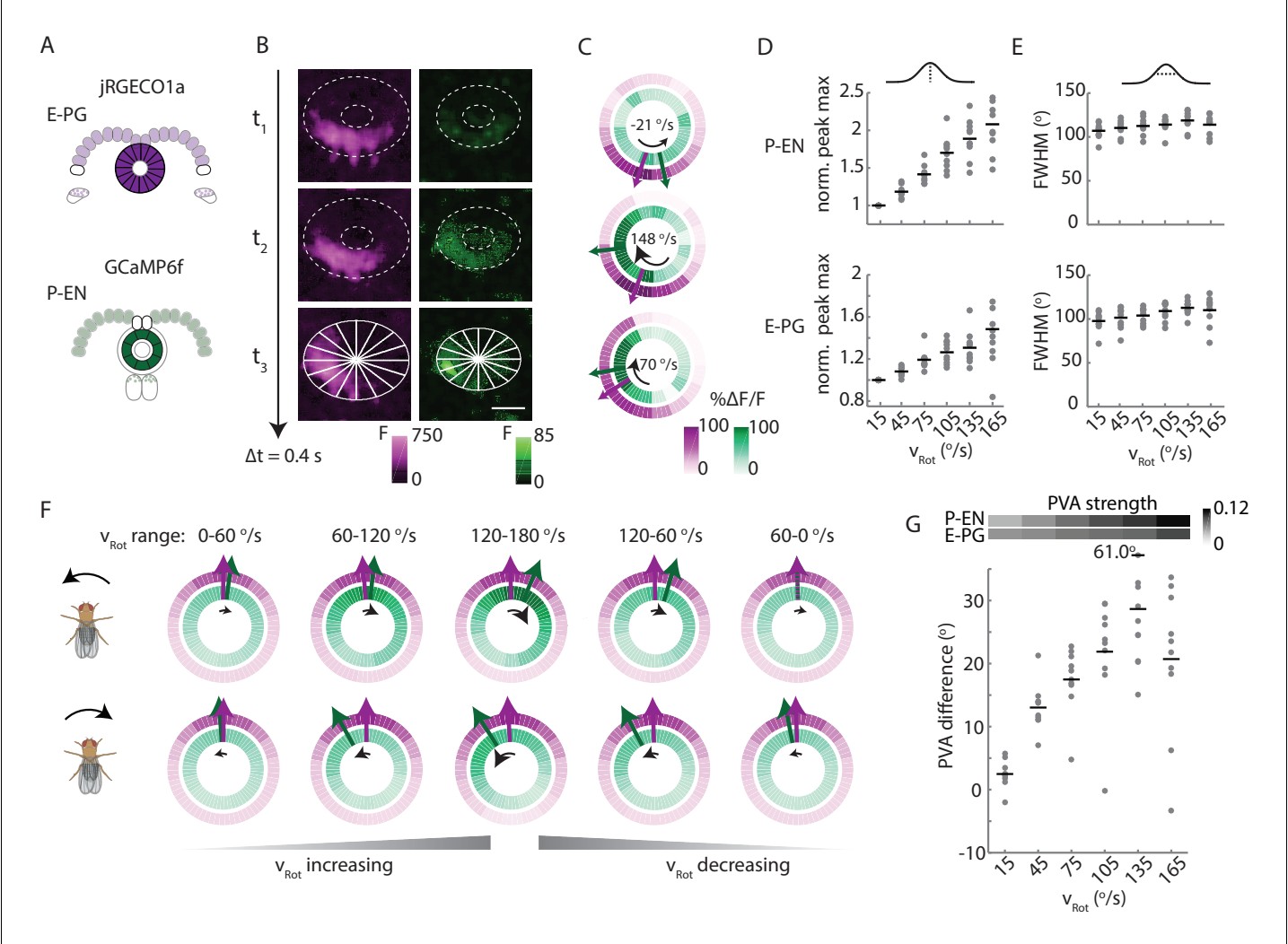

**Figure 9.** P-EN activity leads E-PG activity in the ellipsoid body. (**A**) Overview of a typical two-color calcium imaging experiment targeting the ellipsoid body. jRGECO1a is expressed in E-PG neurons (shown in magenta), and GCaMP6f is expressed in the P-EN neurons (shown in green). (**B**) Maximum intensity projections of fluorescence from simultaneous imaging of the E-PG and P-EN neural populations. Time points are 0.4 s apart. The bottom images show the regions of interest (solid white lines) and a scale bar of 20 μm. (**C**) Fluorescence transients and PVA for the three successive time points shown in **B**, labeled with the corresponding rotational velocities. The direction of bump movement is shown in the inner black arrows. The colored arrows represent the direction but not the magnitude of the PVA. (**D**) P-EN and E-PG bump amplitude as a function of rotational velocity. The center of each rotational velocity bin is shown. Bin width is 30°/s. Each point represents the mean across trials for one fly and bars show the mean across flies. The values for each fly are normalized to the intensity in the 0–30°/s bin. Statistics for flies with the two populations labeled with the opposite colored indicators are shown in *Figure 9—figure supplement 2*. Normalized P-EN (E-PG) peaks increased at a rate of $7.4 \pm 2.3 \times 10^{-3}$ $(3.1 \pm 1.6 \times 10^{-3})$ s/° for the green indicator and a rate of $2.7 \pm 1.23 \times 10^{-3}$ $(3.0 \pm 2.2 \times 10^{-3})$ s/° for the red indicator. (**E**) P-EN and E-PG bump half widths (FWHM). Each point represents the mean across trials for one fly and bars show the mean across flies. Statistics for flies with the two populations labeled with the opposite colored indicators are shown in *Figure 9—figure supplement 2*. Normalized P-EN (E–PG) widths, obtained by dividing each width by the bump width for the given fly at 0–30°/s, were flat across velocity bins, with slopes of $-0.20 \pm 0.57 \times 10^{-3}$ $(-0.36 \pm 1.3 \times 10^{-3})$ s/° for the green indicator and $0.92 \pm 0.81 \times 10^{-3}$ $(0.48 \pm 1.1 \times 10^{-3})$ s/° for the red indicator. (**F**) Mean fluorescence transients binned by rotational velocity and aligned to the peak of E-PG activity at 12 o'clock across five trials for the fly shown in **A–C**. (**G**) Difference in PVA for mean P-EN and E-PG fluorescence transients across flies and rotational velocity bins. Each point represents the mean across trials for one fly and bars show the mean across flies. (N = 10 flies) The P-EN and E-PG PVA strengths are shown at the top. The offset increase across bins is significant (one-way ANOVA across angular velocities, p=$1.3 \times 10^{-7}$). The PVA offset at 150–180°/s is 20.7 ± 11.7°. If the PVA offset is instead calculated for each individual pair of traces, the offset is 14.6 ± 20.9° The peak offset of the mean traces is 24.8 ± 22.4° and the mean offset of the peak differences of the individual traces is 25.8 ± 9.8°.

The following figure supplements are available for figure 9:

**Figure supplement 1.** Cross sections of binned and aligned E-PG and P-EN activity.

*Figure 9 continued on next page*

*Figure 9 continued*

**Figure supplement 2.** Statistics of P-EN and E-PG activity in the ellipsoid body for flies with the opposite calcium indicator combination.

typical bump full width at half maximum in the protocerebral bridge. Finally, our functional connectivity experiments suggested that E-PG to P-EN connections may involve not just direct excitation, but the likely recruitment of inhibition as well. Thus, the simple conceptual model could not sufficiently explain the circuit dynamics we observed.

We therefore designed and simulated a firing rate model that built on the known facts about the circuit's topology. We separated the P-EN neurons into two subpopulations of nine neurons each (one neuron per protocerebral bridge glomerulus, see Materials and methods), one population for neurons with dendritic projections on the right side of the protocerebral bridge, and another for neurons with dendritic projections on the left side. 54 E-PG neurons (three neurons per protocerebral bridge glomerulus) locally excite P-EN neurons, with each P-EN neuron innervating one glomerulus of the protocerebral bridge. In return, P-EN neurons excite E-PG neurons in the ellipsoid body, but with a shift in either the clockwise or counterclockwise direction, depending on whether they belong to the left or the right subpopulation. Further, we stipulated that the E-PG neurons uniformly and strongly inhibit P-EN neurons (all connections shown in *Figure 10A*; see *Figure 10—figure supplement 1A* for the connectivity matrix; see Discussion for possible inhibitory pathways). Such inhibition was a prerequisite to obtain a stationary bump of activity and a near-linear integration of velocity inputs (*Figure 10—figure supplement 1B*; see below). Finally, turns were initiated by an external input to the model that uniformly excites either the right or left P-EN subpopulations (*Figure 10A*, bottom).

This rate model, which is reminiscent of past models of mammalian head direction cells (*Skaggs et al., 1995*; *Xie et al., 2002*; *Zhang, 1996*), albeit with key differences in connectivity, captured the essence of much of what we observed in our experiments. Simulated firing rate curves of P-ENs in response to rotational velocity input were sigmoidal with an inflexion point at 0°/s, matching our experimental observations (compare *Figure 10—figure supplement 1C* and *Figure 4C*). Further, the simulations indicated that the circuit functions as a ring attractor, enforcing a unique bump across the E-PG population, consistent with previous experimental results (*Seelig and Jayaraman, 2015*, *Kim et al., 2017*) (*Figure 10B,C*). Our model does not feature an attractor network within the E-PG population. Rather, E-PG activity is driven entirely by P-EN neurons and the persistent bump of activity arises from a feedback loop between these two neural populations. As a result, the activity and bump amplitudes of

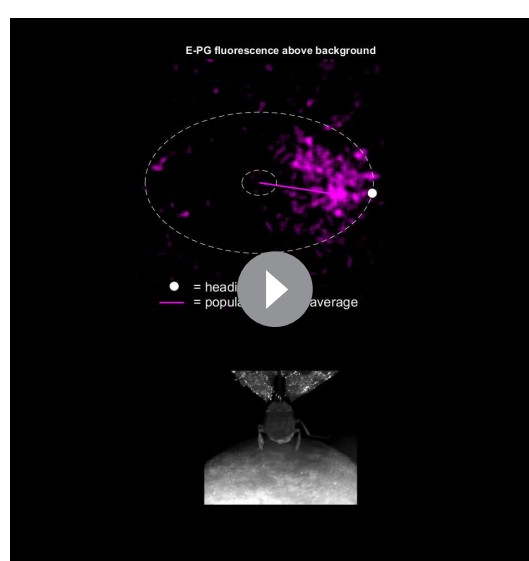

**Video 2.** P-EN bump leads the E-PG bump in the ellipsoid body. Example of two-color calcium imaging in the ellipsoid body of a fly walking on a ball. Part 1. (Top) Red channel signal from the E-PG neurons. The outline of the E-PG arborizations in the ellipsoid body is marked with dotted white lines. The heading of the fly, as read out from the ball position, is shown by the white circle. The population vector average of the activity is labeled by the magenta line. (Bottom) Video of the fly on the ball. Part 2. (Top) Green channel signal from the P-EN neurons. The outline of the P-EN arborizations in the ellipsoid body is marked with dotted white lines. The amplitude of the rotational velocity is indicated by the white bar at right. The population vector average is labeled with the green line. (Bottom) Video of the fly on the ball. Part 3. (Top) Combined red and green activity and PVA of both the E-PG and P-EN neurons. (Bottom) Video of the fly on the ball.

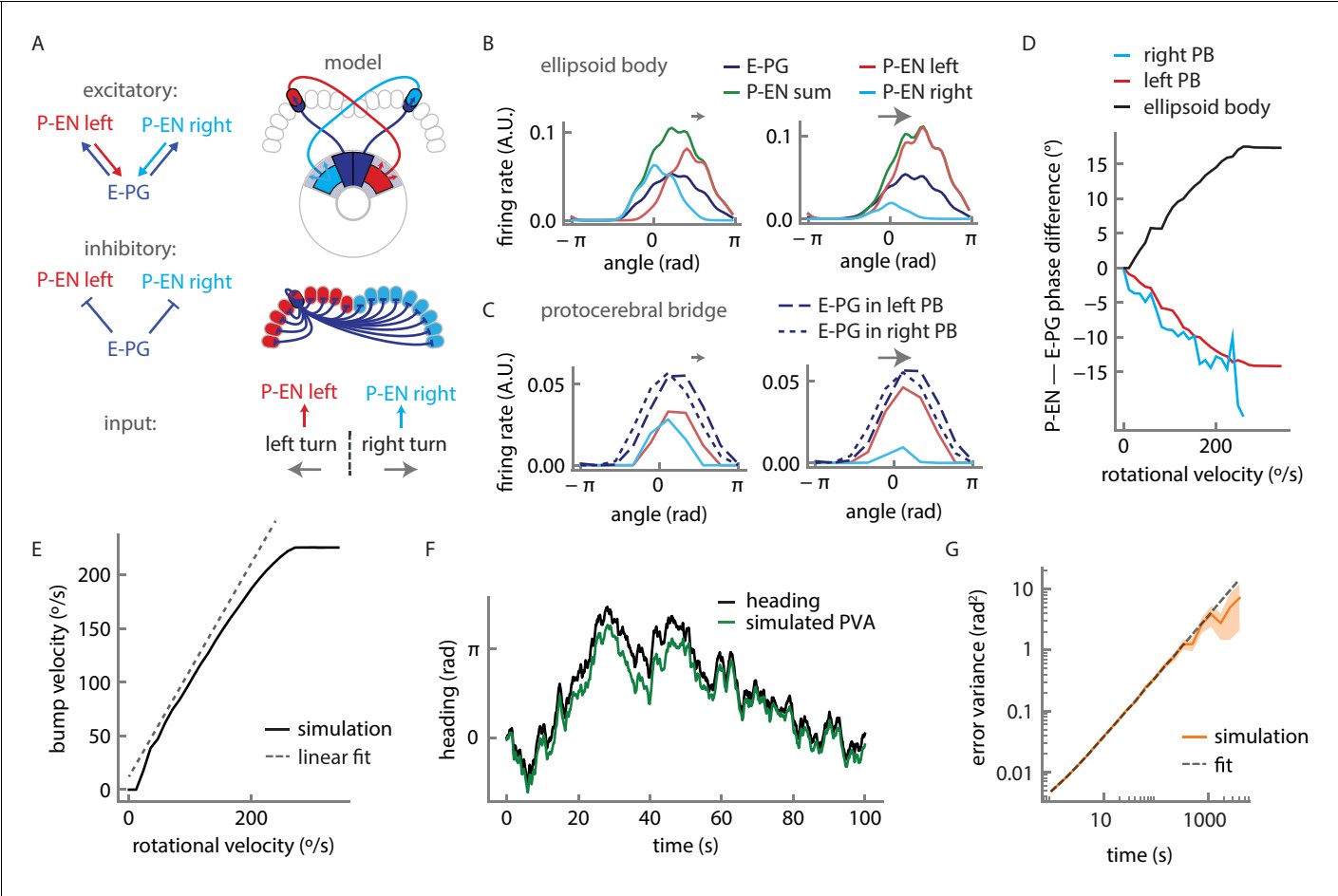

**Figure 10.** Firing rate model for a circuit mechanism displaying persistent localized activity and angular velocity integration. (**A**) Schematic of effective excitatory (top) and inhibitory (middle) connectivity assumed in the firing rate model and external inputs to the P-EN populations (bottom). Note the anatomical shift in the ellipsoid body between E-PG and P-EN neurons relative to their protocerebral bridge connections. We assume one P-EN and three E-PG neurons per protocerebral bridge glomerulus. (**B**) Activity of E-PG and P-EN neurons in the ellipsoid body for counterclockwise turns at low (left, 35°/s) and high—that is, close to saturation— (right, 190°/s) angular velocities. (**C**) Activity of E-PG and P-EN neurons in the protocerebral bridge at low (left) and high (right) angular velocities for a snapshot in time. The velocities are the same as in **B**. (**D**) PVA difference between P-EN and E-PG bumps in the ellipsoid body (black) or protocerebral bridge (red, blue) for different angular velocities for counterclockwise turns. (**E**) E-PG bump velocity as a function of the fly's rotational velocity. The bump velocity displays saturation at high velocities. A linear fit of slope one around the origin is also displayed (upward shifted for display purposes). Rotational velocities along this line will be reliably integrated. (**F**) Simulated PVA of the E-PG population as a function of time for a time varying rotational velocity input (see *Figure 10—figure supplement 2* for a description of the input). (**G**) Evolution of the estimator of the error variance between the velocity input and the simulated PVA. Beyond 10 s, the statistics of the discrepancy follow a diffusion equation with a diffusion coefficient of $1.82 \times 10^{-3}$ rad²/s (see Materials and methods for a description of the fitting procedure). The shaded area indicates the standard deviation of the estimator.

The following figure supplements are available for figure 10:

**Figure supplement 1.** Model connectivity and performance.

**Figure supplement 2.** Angular velocity statistics.

both populations are sensitive to velocity input, as observed experimentally (*Figure 10—figure supplement 1C,D*).

The firing rate model also produced offsets in the activity of the E-PG and P-EN neurons in the ellipsoid body and in the protocerebral bridge, qualitatively matching our experimental observations. In the simulations, the offsets are due to the neural time constants (see

Materials and methods); the bump in the P-EN population is not updated immediately after the E-PG bump shifts. We thus observe a phase difference in the bridge, with the P-EN bumps lagging the E-PG bumps, as observed experimentally (*Figure 8G*). This lead-lag relationship was reversed in the ellipsoid body, where the P-EN population drives the E-PG population. The neural constants are expected to be largely independent of the bump's velocity. Consequently, when the E-PG bump in the protocerebral bridge, for instance, traveled further in a given time window (as during a fast turn), the P-EN bump lag and phase difference increased, consistent with experimental observations (*Figure 9F*, *Figure 10D*).

For an animal to reliably track its heading over long periods of time, it must accurately integrate its rotational velocity. Thus, in a robust system, the heading representation should update with an internal rotational velocity that matches the animal's external velocity. Indeed, the architecture of our model ensures a quasi-linear relationship between bump rotation and steady velocity input (*Figure 10E*) up to an inherent saturating angular velocity. This linearity mainly depends on an appropriate balance of local E-PG to P-EN excitation, and inhibition onto the P-EN neurons (*Figure 10—figure supplement 1B*). When we drove the P-ENs with a simulated but realistic time varying velocity input (*Figure 10—figure supplement 2*), we saw that the bump in the model closely followed the integrated external velocity, reliably tracking the animal's heading (see *Figure 10F*, *Figure 10—figure supplement 1E,F*). The errors that did arise were due to inaccurate integration of small input velocities. The limited number of P-EN neurons in the model (nine per subpopulation) caused the bump to 'stick' at individual P-EN neurons until the input velocity provides enough drive to free it, at values greater than 15°/s. The integration error, the difference between the bump position and the integrated velocity input, followed a distribution centered on zero, with a variance that increased over time (see Materials and methods). For short times (on the order of 10 s or less), the variance of the distribution was small, that is, the error between the animal's heading and the animal's representation of its heading was likely to be small (*Figure 10—figure supplement 1E*). This error was influenced by the neural time delays of the angular integration circuit (*Figure 10—figure supplement 1F*). For longer time scales, the error distribution broadened (*Figure 10G*) so that the variance increased linearly, a characteristic feature of a diffusion process (that is, the heading representation diffused away from the true heading over time). Overall, the linearity of the heading response to input and the small value of the diffusion coefficient suggest that this simple model of only three neural populations and E-PG to P-EN inhibition, which may be mediated by an additional and as-yet-uncharacterized neural population, can not only maintain a bump of activity, but can accurately update that bump position to reflect the animal's heading.

## Blocking the P-EN synaptic output alters the E-PG activity

Both the conceptual model (*Figure 1H*) and the quantitative model (*Figure 10*) rely on P-EN input to the E-PG neurons to maintain and move the bump. To assess the impact of the loss of P-EN input on E-PG bump strength and movement, we imaged E-PG activity while impairing P-EN synaptic transmission. To that end, we used shi[TS], a temperature-sensitive mutation of the *Drosophila* gene encoding a dynamin orthologue, which blocks vesicle endocytosis at elevated temperatures (*Kitamoto, 2001*). We compared E-PG activity between control flies expressing shi[TS] in an empty promoter Gal4 and flies expressing shi[TS] in two different P-EN lines, R37F06 and VT008135. For each line, we compared the activity at room temperature (permitting synaptic transmission) to the activity at the restrictive temperature of 31°C, where vesicle reuptake and, thus, synaptic transmission should be blocked (*Figure 11*).

We first checked that the flies' behavior was consistent across genotypes. Indeed, the walking statistics were similar at a given temperature across GAL4 lines (see, for example, the distributions for two flies at 31°C in *Figure 11Ai*), though we consistently observed a strong increase in forward walking at the higher temperature (*Figure 11Aii*). In experiments with control flies, we found that this increased forward velocity reliably led to a high-amplitude E-PG activity bump at the higher temperature (*Figure 11B*, top). As expected, the bump position slowly drifted away from the fly's heading in darkness, but otherwise tracked the fly's rotations (*Figure 11B*, bottom). In contrast, in flies that expressed shi[TS] in P-EN neurons, the bump was significantly weaker and more variable (*Figure 11C*), although these flies walked just as actively at the higher temperature as control flies.

We next quantified the effects of blocking P-EN neurons on E-PG bump amplitude. At room temperature, the distribution of E-PG bump amplitude across trials was similar for the control flies and

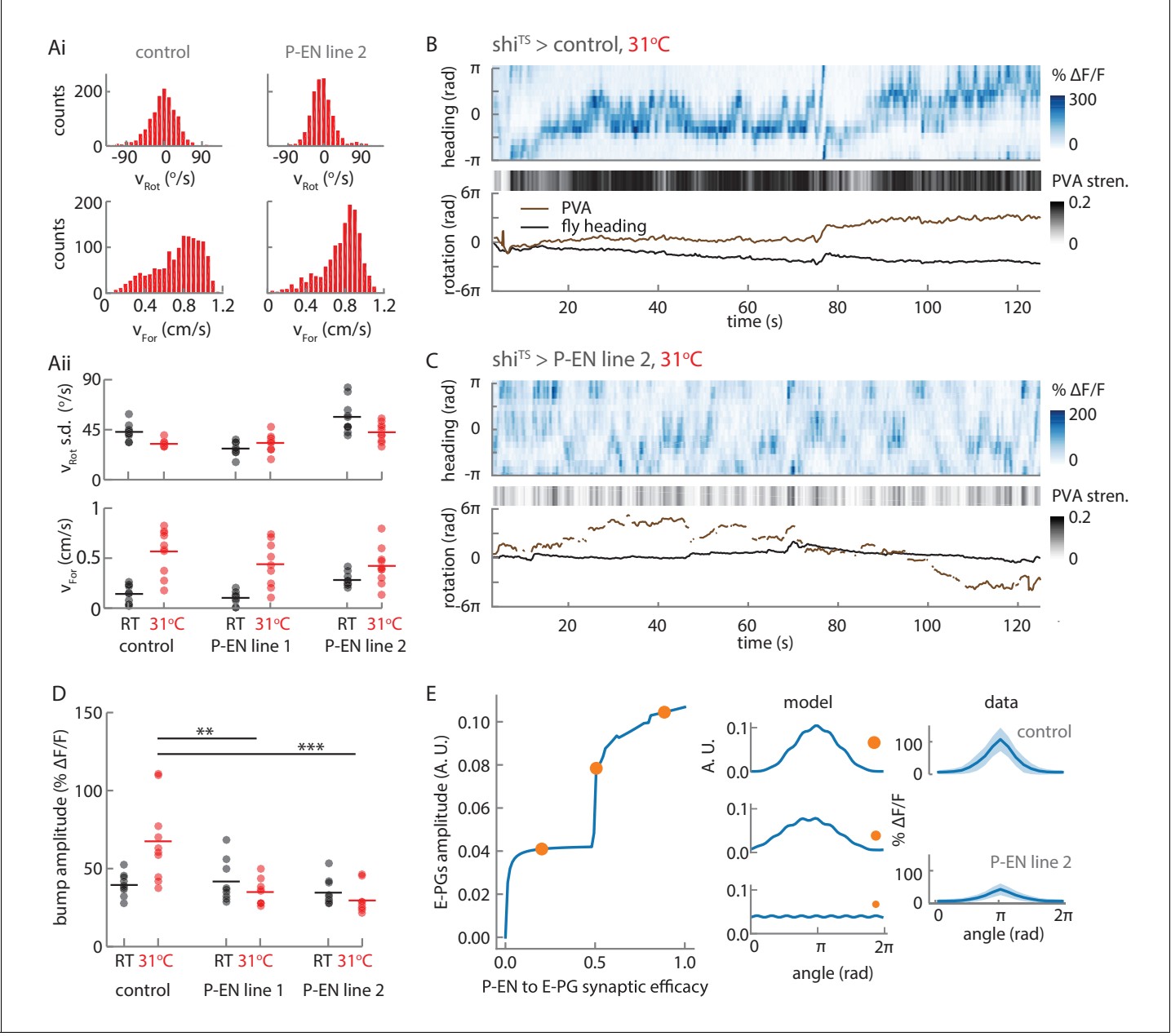

**Figure 11.** Synaptic block of P-EN neurons disrupts E-PG activity. (**A**) Walking statistics for flies across temperatures and across genotypes. (**Ai**) Example rotational velocity (top) and forward velocity (bottom) distributions for single trials for control (left, shi[TS] expressed in Empty Gal4) and P-EN blocked (right, shi[TS] expressed in the P-EN containing line R37F06, herein referred to as P-EN line 2). (**Aii**) Walking statistics at room temperature (RT, black) and at 31°C (red) for the control line and for two lines that express shi[TS] in the P-EN neurons. P-EN line one is GAL4-VT008135. The standard deviation of the rotational velocity is plotted at top, while the median of the forward velocity is shown at bottom. Lines show the means of all values. (**B**) Recordings at 31°C for one fly with shi[TS] expressed in empty Gal4. (Top) E-PG activity recorded in the ellipsoid body. (Center) The E-PG PVA strength. (Bottom) E-PG PVA and fly heading. The PVA is only shown when the PVA strength exceeds 0.025. (**C**) Same as in A, but now for a fly with shi[TS] expressed in the P-EN neurons (P-EN line 2). The PVA strength is lower and the bump movement is sporadic. The PVA is only shown when the PVA strength exceeds 0.025. (**D**) Mean bump amplitude at RT and at 31°C for all flies and fly lines (N = 10 for the control, N = 9 for P-EN line 1, N = 10 for P-EN line 2, bootstrapping gives p=0.0056 (**) and p=6.0×10⁻⁴ (***)). (**E**) Comparison of the firing rate model predictions and the shi[TS] data. (Left) E-PG bump amplitude as a function of the P-EN to E-PG synaptic efficacy for the firing rate model. (Center) The predicted ellipsoid body activity for three points along the curve that are marked with orange circles. (Right) The mean and standard deviation of the ellipsoid body activity across trials for the control fly shown in **B** and the shi[TS] > P-EN line 2 fly shown in **C**. The synaptic efficacy for the shi[TS] fly, likely less than 1, is unknown.

The following figure supplement is available for figure 11:

*Figure 11 continued*

**Figure supplement 1.** PVA behavior during turns under synaptic block.

for flies that expressed shi[TS] in the P-EN neurons (*Figure 11D*). At 31°C, however, when the flies became more active and increased their forward velocity, the amplitude distribution of the control flies shifted to higher values. In contrast, in the shi[TS] P-EN flies, where this activity increase was accompanied by a synaptic block in P-EN neurons, the bump amplitude distribution remained relatively unchanged, or even decreased (*Figure 11D*).

We looked to see if this reduced amplitude could be explained by our firing rate model. The model demonstrated that two recurrent E-PG/P-EN loops are sufficient to maintain and shift a bump of activity (*Figure 10*). If connections between the model's loops are broken, the bump would be expected to disappear. To explore the consequences of weakening the P-EN to E-PG connections instead of breaking them entirely, we continuously tuned the weights down to 0 from their initial value (thereby modifying the P-EN to E-PG synaptic efficacy) and observed a range of simulated activity profiles in the ellipsoid body (*Figure 11E*, *Figure 11—figure supplement 1C*). When the synaptic efficacy was close to 0, activity in the network was relatively uniform and did not localize into a bump. At higher P-EN to E-PG synaptic efficacies, network activity localized into a bump, and its amplitude increased as the efficacy of the synaptic connection between P-EN and E-PG neurons increased. This critical role for the P-EN-to-E-PG connection in maintaining bump strength was consistent with the reduction in bump amplitude that we observed experimentally across flies and genotypes when P-EN output was impaired (*Figure 11D*).

Finally, we examined the reliability of the E-PG bump in tracking turns across flies and across conditions. Individual turns were defined as continuous periods where the rotational speed exceeded 15°/s. To guarantee that we could reliably track the bump across these turns, all turns with a PVA amplitude below a set signal-to-noise threshold were excluded from this analysis (*Figure 11—figure supplement 1A*, see Materials and methods). We then compared the change in PVA position to the change in the fly's heading over the course of these turns (*Figure 11—figure supplement 1B*). We fit the data with a simple linear model and extracted $R^2$ values to gauge how reliably the PVA tracked turns (*Figure 11—figure supplement 1D,E*). These $R^2$ values were, on average, lower for the P-EN lines than for the control line, suggesting that disrupting the P-EN synaptic output affects the reliability of the E-PG bump to track turns. However, while our conceptual model and our simulation suggest that the E-PG bump should move less when P-EN output is reduced (*Figure 11—figure supplement 1C*), we instead observed that the E-PG bump often drifted at the restrictive temperature in the shi[TS] P-EN flies (see Discussion), as can be seen in part from the wide range of PVA changes seen during turns (*Figure 11—figure supplement 1B*).

Overall, although our shi[TS] results could not clarify how exclusive a role the P-EN neurons play in moving the E-PG bump in darkness, they revealed the importance of P-EN activity for maintaining the E-PG bump's strength and stability, consistent with the assumptions of our model.

## Discussion

A stable internal representation of heading is fundamental to successful navigation. Neurons that maintain such a representation in darkness have been reported across various species (*Knierim and Zhang, 2012*; *Seelig and Jayaraman, 2015*; *Taube et al., 1990*; *Varga and Ritzmann, 2016*). Several computational models have been proposed to explain how a population representation of heading might be updated using angular velocity signals from different neural populations, but identifying connections between neurons that carry and integrate these disparate signals has been challenging in mammals (*Knierim and Zhang, 2012*). Here, we took advantage of the small size, strong topography and well-described anatomy and cell types of the fly central complex to identify a candidate neuron population, P-ENs, which carry angular velocity signals. We used cell-type-specific genetic tools to perform electrophysiological recordings from single P-EN neurons and two-photon calcium imaging from entire populations of both P-ENs and the previously described 'compass neurons' (E-PGs) in head-fixed walking flies to demonstrate how these neurons together create an elegant circuit mechanism to update a heading representation when the fly turns in darkness. The

circuit motif underlying this mechanism (see cartoon animation of this in *Video 1*) shares some characteristics with past conceptual models of head-direction cell function (*Hartmann and Wehner, 1995*; *Skaggs et al., 1995*; *Xie et al., 2002*).

The rate model we implemented was able to capture the essence of the observed network activity, reproducing physiological activity in response to an input that is specific to one side of the protocerebral bridge, but uniform otherwise. This suggests a level of control over moving the activity bump that is quite simple to implement in neural circuitry. In addition, our model is agnostic to the type of input that is needed to rotate the bump. It does, however, require inputs that are activated when the fly turns, with a strength proportional to the strength of the turn, and that such inputs preferentially innervate one hemisphere to create a mirror-symmetry in the system. This description anatomically matches at least one known cell type: PB$_{G1/2-9}$.b-SPSi.s (*Wolff et al., 2015*). The model also requires inhibition to maintain a stationary bump and linear velocity integration. The widely arborizing and glutamatergic PB18.s-Gx$\Delta$7Gy.b neurons (*Daniels et al., 2008*; *Wolff et al., 2015*) may provide such large-scale inhibition onto the P-EN neurons.

Some discrepancies remain between our proposed model and the experimental evidence we present here. Our model assumes only one P-EN neuron per protocerebral bridge glomerulus (*Wolff et al., 2015*), which puts a strong constraint on the angular velocity integration properties of the circuit. In particular, although the circuit displays linear velocity integration within the typical range of angular velocities, the activity bump gets 'stuck' at individual P-EN neurons for small turns (*Figure 10E*). That is, when the fly turns slowly, the corresponding small inputs to the circuit do not trigger bump movements. We did not observe such bump dynamics in our imaging experiments, indicating that other, unexplored factors may help smooth bump movement in the actual circuit. Noise in the circuit, potential gap junctions and dendro-dendritic connections within and between E-PG and P-EN neurons, as well as the activity of other cell types in the circuit, such as the PB$_{G1-8}$.s-EBt.b-D/Vgall.b neurons (*Wolff et al., 2015*), may all play a role in smoothing bump movement. These factors may also contribute to differences in bump shape and width between the model and experimental data. Further, our model suggests that E-PG activity is directly passed to the P-EN neurons in the protocerebral bridge, possibly with some anatomical offset and modulation through inhibition. Indeed, we observed almost coincident bumps of activity in the bridge for the two cell types. However, while functional connectivity showed a clear connection from the P-EN neurons to the E-PG neurons, our connectivity, electrophysiology, and imaging results suggested that the E-PG to P-EN connection might be more indirect and also recruit inhibition. In the functional connectivity experiments, very strong activation of the E-PG population reliably excited the P-EN neurons, but weaker excitation evoked a variety of responses (*Figure 7D*). Our electrophysiological recordings also revealed an unanticipated complexity in the tuning of the P-ENs' membrane potential. Membrane potential tuning curves generally showed a peak at the same heading as the spike rate tuning curves, but also a pronounced trough about 150° distant from that peak. That trough, likely a result of inhibition in the circuit, was not always evident in the spike rate tuning (see *Figure 5C* for an exception). Finally, in two-color imaging, we observed offsets of up to one glomerulus between the E-PG and P-EN activity on the ipsilateral side of the bridge, and unexpected P-EN activity on the contralateral side, also offset from the E-PG activity (discussed below). These results were consistent for both color indicator pairings, as well as in experiments involving a second driver line (VT008135, see Materials and methods for further details, *Figure 8—figure supplement 2*), suggesting that the effects are not merely an artifact of indicator kinetics or co-expression in another population of neurons. We noted that during slow rotations, when P-EN activity is low and the E-PG bump is weak, these offsets decreased and, depending on the driver line used, also differed between the ipsi- and contralateral side during a turn (compare *Figure 8G* and *Figure 8—figure supplement 2*). We take this as indications that the connectivity between the E-PG and P-EN neurons in the protocerebral bridge may be partly indirect. Future studies will address how excitatory and inhibitory connectivity between these populations and others shape the circuit's compass function.

Still uncertain is whether an activity bump can be independently sustained in the P-EN and E-PG populations, or in the left vs. right P-EN subpopulations. The connections from P-ENs to E-PGs may be the substrate that sustains the maintenance of E-PG bump position in the ellipsoid body in the absence of both visual and self-motion cues (*Seelig and Jayaraman, 2015*), as in our model (*Figure 11E*). The significant reduction we saw in E-PG bump amplitude and PVA strength when synaptic transmission from P-ENs was blocked (*Figure 11*) is supportive of such an idea. Whether E-PG

input is similarly essential to the maintenance of P-EN bump strength is less clear, but P-EN heading tuning hints at a dependence on E-PG input. On the other hand, appropriate local connections between nearby neurons either in the ellipsoid body or in the protocerebral bridge may allow bumps of activity to be independently sustained in the E-PG neurons. Signs of such internal connections come, for example, from evidence of presynaptic specializations of E-PGs in the ellipsoid body (see synaptotagmin labeling in *Figure 7B*). Bump persistence could also be achieved through long time-scale cellular biophysics (*Yoshida and Hasselmo, 2009*). Future experiments and electron micros-copy-based circuit reconstruction efforts should provide stronger constraints on the space of possi-ble models, and clarify the functional and behavioral relevance of the actual circuit structure.

## Comparing results from electrophysiological and calcium imaging experiments

For a circuit mechanism in which phase relationships and conjunctive coding are important, calcium imaging may seem an unreliable arbiter of truth. Somatic single cell recordings, on the other hand, can be hard to interpret given the intricate projection patterns of fly neurons and the compartmen-talization of information processing that this can produce (*Yang et al., 2016*). However, we found the results from our calcium imaging and electrophysiology experiments in P-EN neurons to be in broad agreement. The electrical signature of P-EN responses to angular and forward velocity mir-rored what we saw with calcium imaging in the noduli. The measured width of a single P-EN neu-ron's receptive field (~60°) was lower than that observed with calcium imaging (~110°, *Figure 9* and *Figure 9—figure supplement 2*), but this may arise from the slow decay kinetics of calcium indi-cators. One inconsistency between results, however, related to the imaging of neural activity in the protocerebral bridge. Based on imaging in the noduli (*Figure 2*) and electrophysiology (*Figure 4* and *Figure 5*), we expected that turns in one direction would evoke a steady decrease in activity on the other (contralateral) side of the bridge with increasing rotational velocity. Instead, imaging in the bridge showed a mild increase in activity at higher velocities (*Figure 8*), albeit while preserving the expected asymmetry between the ipsi- and contralateral side (*Figure 8—figure supplement 2*). We hypothesize that this calcium signal might represent synaptic inputs to the P-ENs more than their spiking activity.

## The role of P-ENs in the maintenance, stabilization and movement of the E-PG bump

Blocking P-EN output using shi[TS] had two effects on the E-PG bump: Its amplitude was reduced and its position sometimes changed dramatically during small turns (visible as an increase in vari-ability and low $R^2$ for the correlation of changes in heading versus PVA in *Figure 11—figure sup-plement 1B,D*). The bump amplitude decrease in shi[TS] flies at high temperature can be readily explained by the reduction in synaptic input to the E-PGs — indeed, in our firing rate model (*Fig-ure 10*) P-EN input is essential to the maintenance of the E-PG bump. Several factors may explain why the E-PG bump did not completely disappear during this manipulation. First, cell-intrinsic prop-erties of the E-PG neurons may contribute to the persistence of activity in those neurons even in the absence of external input. Second, the shi[TS] block may have been incomplete (*Thum et al., 2006*), meaning that there was sufficient P-EN drive even at high temperatures to keep the E-PG bump alive. Third, we cannot rule out the possibility of gap junctions between P-EN and E-PG neu-rons, which our experiments would not block. Finally, as mentioned above, other neuron types, such as the PB$_{G1-8}$.s-EBt.b-D/Vgall.b (*Wolff et al., 2015*), may also provide synaptic input to the E-PG population in the ellipsoid body. Some of these possibilities have been suggested in a recent study that used the anatomy of protocerebral bridge neurons to create a spiking model that gener-ates ring attractor dynamics (*Kakaria and de Bivort, 2017*).

Further, our conceptual and firing rate models would imply that if the P-EN to E-PG connections were entirely removed, E-PG activity would be unable to follow the fly's turns. However, the E-PG activity does still track the fly's turns at high temperature, when the P-EN synaptic output should be blocked. This may, once again, be the result of an incomplete block. We speculate that one reason that the E-PG bump makes large movements across the ellipsoid body even during small turns is that a reduction in bump amplitude destabilizes the compass representation. Thus, fluctuations in the activity of the E-PGs elsewhere in the ellipsoid body may exert a greater influence on the

movements of the bump than under normal conditions, when activity in distant E-PGs is likely to be suppressed (*Kim et al., 2017*). Yet another possibility that could explain the bump's movements is raised by a parallel study, which provides further evidence for P-ENs serving a role in angular integration and describes a second subtype of P-EN neurons that likely also influences the position of the E-PG bump (*Green et al., 2017*).

## The role of the compass representation in the fly's behavior

The coordinated activity of the E-PG population and its control by the P-EN population when the fly turns are strongly evocative of a compass. The animal could, in principle, use such a neural compass to tether its actions to local landmarks or other sensory cues during navigation, and maintain its bearings in the temporary absence of such cues (*Neuser et al., 2008*). Consistent with this idea, the PVA computed with E-PG population activity tracks the fly's heading quite accurately even in darkness. However, we do not yet know how downstream circuits read out E-PG population activity. Thus, although the PVA metric is a useful representation of E-PG compass-like activity, whether downstream circuits perform similar computations to extract the fly's heading is unclear. Further, we derive the PVA by combining the strength and angular position of activity in the E-PG population. Although both these features of E-PG activity likely influence downstream neurons, their specific influence on such neurons will depend on the precise connectivity of the circuit, something that a combination of functional connectivity studies and electron microscopy may reveal in time. Although there is considerable evidence across insects suggesting that CX neurons influence action initiation and turning movements (*Bender et al., 2010*; *Martin et al., 2015*), the connection of E-PG and P-EN neurons to the largely unidentified class of CX neurons that drive behavioral decisions is as yet unclear.

## Other elements of the compass system

We have focused here on the effects of self-motion cues on bump movement, to which end most of our experiments were conducted with flies walking in the dark. However, E-PG activity is strongly influenced by visual cues, as evidenced by the fact that cue jumps can reset the bump position (*Seelig and Jayaraman, 2015*; *Kim et al., 2017*). The angular velocity representation of P-EN neurons, by contrast, seemed unaffected by the presence of closed loop visual feedback (*Figure 4—figure supplement 1*). Thus, while we have suggested a circuit mechanism for updating heading representation in the dark using self-motion signals, we anticipate that strong sensory inputs, including those from visual cues, control updating in other circumstances. For example, we have previously observed that the ring neurons retinotopically respond to visual cues (*Seelig and Jayaraman, 2013*). As the putative ring neuron axons arborize in the ellipsoid body along with the E-PG dendrites, it may be possible for them to convey visual information to the E-PG neurons, influencing the movement of the bump of activity. Further, we suggested above that the E-PG to P-EN connection in the protocerebral bridge may be indirect and recruit sources of inhibition. There exist a few classes of bridge interneurons, the so-called $PB_{G6-8}.s_{G9}.b$, PB18.s-GxΔ7Gy.b and PB18.s-9i1i8c.b neurons (*Wolff et al., 2015*), which may serve as intermediaries in E-PG to P-EN connections (*Kakaria and de Bivort, 2017*). Future studies should help clarify their role in the compass network.

## Angular velocity signals and heading representation in the fly and other animals

Fly E-PG neurons share several characteristics with mammalian head direction cells. Both head direction cells and E-PG neurons maintain one stable bump of activity and both track the animal's heading in darkness, a feature that is well described by appropriately wired ring attractor models. Rodents that are deprived of proprioceptive and motor efference signals, as in passive transport experiments (*Stackman et al., 2003*), show impaired heading representation. To update their heading in darkness, head direction cells in rodent thalamic nuclei and post-subiculum are thought to depend on angular velocity input from the vestibular system, mediated by the dorsal tegmental nucleus (*Bassett and Taube, 2001*). Although 75% of neurons in this region were found to encode angular head velocity, only about a third of those did so in the mirror-symmetric, turn-direction-selective fashion of *Drosophila* P-EN neurons that we describe here.

Individual P-EN neurons were deterministic in their left-right mirror-symmetric rotation tuning, but diverse in the range of rotational velocities that their tuning curves spanned. Indeed, the measured bandwidth of individual P-ENs ranged anywhere between 30 and 270°/s. This diversity may reflect the diversity of tuning of the three to four P-EN neurons that innervate each protocerebral bridge glomerulus (estimated from cell body counts [*Wolff et al., 2015*]). Such a range of sensitivities and bandwidths would permit a more precise tracking of the flies' turns across a wide range of rotational velocities.

The origins of angular velocity responses in P-ENs are as yet unclear, but these responses show a latency relative to the fly's turning movements that we estimated to be ~150 ms, suggesting that they arise from proprioception rather than motor efference. Anatomically, both the two halves of the protocerebral bridge as well as the two noduli are mirror-symmetric structures innervated by a number of neuron types in a lateralized manner (for example, PB$_{G2-9}$.b-IB.s.SPS.s [*Wolff et al., 2015*]), making them likely candidates for receiving such rotation-tuned input. In the cockroach, neurons encoding angular as well as forward velocity have been recorded in the fan-shaped body (*Guo and Ritzmann, 2013*; *Martin et al., 2015*), a substructure of the central complex that is evolutionarily conserved in flies. Of note, only one of the forty turn responsive neurons in the latter study showed bidirectional modulation, with excitation for turns in the preferred direction and inhibition for turns the other way, a hallmark of the P-EN neurons. These studies, which relied on extracellular recordings and did not identify cell types, found that changes in spike rate regularly preceded locomotor changes instead of tracking them as we found for the fly P-ENs. If we assume that neurons of the type recorded in the cockroach also exist in the fly, it is not yet clear whether the P-EN/E-PG compass network that we describe here exploit advance information about expected changes in angular velocity.

## Structure-function relationships in central brain circuits

A striking aspect of the fly compass system is its structural symmetry. Mirror symmetry is a prominent feature of the anatomical layout of the protocerebral bridge. The developmental origins of the anatomical positions of central complex neurons have been the focus of numerous studies (recently, for example, *Boyan and Liu, 2016*). However, although the two sides of the protocerebral bridge and the noduli are tuned to rotations in opposite directions, maintaining symmetry at the large scale, the activity of the E-PGs and P-ENs at the scale of bridge glomeruli breaks this symmetry. During a turn, bumps of activity propagate through the left and right sides of the bridge in parallel, in a manner reminiscent of windshield wipers, rather than obeying mirror symmetry. This pattern of activity, together with the connectivity of protocerebral bridge glomeruli and ellipsoid body sectors (see schematic in *Figure 1G*) ensures that the E-PG bump moves smoothly around the ellipsoid body when the fly turns.

More broadly, topographical organization is a striking feature of many sensory circuits, but structure often follows computational function in neural circuits in the central brain as well. The feedforward pathways to and from the Mauthner cell make clear these neurons role in rapid escape behavior (*Hale et al., 2016*), and the parallel delay loops of the barn owl auditory system and the electric fish point to their comparative roles in localizing prey (*Konishi, 2006*). The anatomical shift of the P-EN neurons with respect to the E-PG neurons (*Wolff et al., 2015*) provided an immediate clue to a potential structure/function relationship, that of a mechanism for shifting the bump of E-PG activity to update their internal representation of heading. The fact that topography often matches topology in the small fly brain makes the system ideal for the identification of circuit mechanisms underlying complex computations. Only time —and perhaps large scale circuit reconstruction efforts (*Denk et al., 2012*)— will tell whether such network motifs are also present, but perhaps better hidden, in the more distributed circuits of much larger brains.

## Materials and methods

### Fly stocks
Calcium imaging (7–10 days old females)

1. UAS-jRGECO1a/LexA-R37F06;LexAop-GCaMP6f/GAL4-R60D05

2. UAS-jRGECO1a/LexA-R60D05;LexAop-GCaMP6f/GAL4-R37F06

## Visually guided patch-clamp recordings (1–2 days old females)

1. GAL4-R37F06/UAS-GFP
2. LexAop-mCD8-GFP;LexA-R37F06;UAS-Jaws-mCherry,GAL4-R55G08
3. LexAop-mCD8-GFP;LexA-R37F06;UAS-Jaws-mCherry,GAL4-R76E11
4. LexA-R37F06;UAS-CsChrimson-mCherry,LexAop-GFP/GAL4-R55G08
5. LexA-R37F06;UAS-CsChrimson,LexAop-GFP/GAL4-R55G08
6. LexA-R60D05;L4-R37F06/UAS-GCamp6m,LexAop-CsChrimson
7. GAL4-R37F06/UAS-Jaws-GFP

## Functional connectivity experiments (5–8 days old females)

1. LexA-R60D05/+;GAL4-R37F06/UAS-GCaMP6-m in attP2 – LexOp-CsChrimson-tdTomato in VK00005
2. LexA-R60D05/+;GAL4-R37F06/UAS-CsChrimson-mCherry in su(Hw)attP1– LexOp-GCaMP6-m in VK00005

## Polarity studies

1. w, pBPhsFlp2::PEST, tub-FRT-Gal80-FRT; pJFRC-5XUAS-IVS-TLN::smGFP-V5 in su(Hw)attP5 (/CyO); pJFRC225-5XUAS-IVS-myr::smGFP-FLAG in VK00005, pJFRC51-3XUAS-IVS-Syt::smGFP-HA in su(Hw)attP1 (/TM2 or TM6b) (gift of Y. Aso).

## shi$^{TS}$ experiments (3–5 days old females)

1. pJFRC99 UAS-shiTS Su(Hw) attp1/LexA-VT025957, LexAop – Gcamp6f (VK00005)/GAL4-X where X was R37F06, VT008135, or pBDPGal4U (attp2) (empty Gal4 control).

Flies were randomly picked from the food vials for all experiments.

### Visualization of GAL4 expression patterns

Imaging stacks for R60D05 and R37F06 were obtained from the FlyLight imaging database (*Jenett et al., 2012*). As the protocerebral bridge expression was significantly dimmer than the ellipsoid body expression, the volume was divided in two, with the ellipsoid body in the front half and the bridge in the back. The two half volumes were then collapsed into maximum intensity projections, and the contrast and brightness independently varied to best show the individual structures. The two images were then combined and used to generate a third maximum intensity projection, which is shown in *Figure 1*.

### Fly preparation for imaging during walking

Flies were prepared for imaging as described in (*Seelig and Jayaraman, 2015*). Briefly, flies were affixed to a metal shim that held their head and thorax in place while allowing their legs and abdomen to move freely. The fly's head was immersed in saline of the following composition (in mM): NaCl (103), KCl (3), TES (5), trehalose 2 H$_2$O (8), glucose (10), NaHCO$_3$ (26), NaH$_2$PO$_4$ (1), CaCl$_2$ 2 H$_2$O (2.5), MgCl$_2$ × 6 H$_2$O (4). The head capsule was opened and soft tissue removed, exposing the brain. For imaging the noduli or ellipsoid body, the fly's head was positioned so that the top of the head was level. For imaging the protocerebral bridge, the head was pushed down so that the back of the head was at a ~30° angle. The flies were then suspended over a free-floating foam ball and allowed to walk freely while an immersion objective was lowered into the saline. An 8 mm diameter, 92 mg ball was used for the fly treadmill.

### Two-photon calcium imaging

Red and green indicator calcium imaging was performed on a custom two-photon microscope running ScanImage 2015 and similar to the one described in (*Seelig and Jayaraman, 2015*). An 875 nm longpass filter was placed in the incident light path to cut out shorter wavelengths from the laser that otherwise contaminated the red channel. A primary dichroic at 705 nm was used to direct emitted light to the red and green PMTs, and a secondary dichroic at 594 nm split the light to the red and green channels. 50 nm bandpass filters centered around 525 and 542 nm were placed in front of the red channel and a 90 nm wide filter centered at 625 nm was set in front of the green channel.

For five of the flies used for ellipsoid body imaging, a 660 nm primary dichroic was used instead of the 705 nm filter.

All imaging was performed using a 1020 nm, femtosecond pulsed laser at ~20 mW power. For the ellipsoid body and noduli imaging, 256 × 256 pixel volumes were obtained. The volumes were ~50 μm on a side and consisted of 6 Z planes, each spaced 8 μm apart and chosen so as to tile the ellipsoid body and noduli. Volumes were obtained at a rate of 11.4 Hz. For protocerebral bridge imaging, 256 × 256 volumes were again obtained, but they were now ~200 μm on a side and consisted of 6 or 12 Z planes imaged at 11.4 Hz and 6.2 Hz, respectively. An 8 μm spacing was used for the six plane volumes, and a 3 or 4 μm spacing was used for the 12 plane volumes. The ball position was tracked at 200 Hz and synchronized to the calcium imaging using TTL pulses triggered after each frame grab.

Recordings began as soon as the fly was placed on the ball and the software was initialized, usually within five minutes, and never longer than 15 min. If the brain was visibly moving more than ~5 μm or if the fly ceased to move within the first four trials, the experiment was discarded. Otherwise, all trials were analyzed.

## Matching the behavioral and imaging data

The timestamps for each imaging frame were found from the TTL signal, and the behavioral data at the point that corresponded to the end of each volume acquisition was tabulated to create down-sampled behavioral data to match the imaging data. Rotational, translational, and side slip velocities were then calculated by taking the difference of the fly's heading, total translation, or side translation across each time point and dividing them by the time between points. Velocities were also Savitzky-Golay filtered over 11 frames with a third order polynomial to smooth out artifacts due to downsampling.

## Correlating noduli activity and fly velocity

We defined ellipsoidal regions of interest over each nodulus, calculated $\Delta F/Fs$, and then computed the difference in activity between the two sides. Rotational velocities were convolved with the function:

$$e^{-t/\tau_{on}} - e^{-t/\tau_{off}},$$

using time constants for GCaMP6f obtained in response to 10 action potentials in dissociated mammalian neurons: $\tau_{on}$ = 0.13 s and $\tau_{off}$ = 0.63 s (*Chen et al., 2013*). The convolved velocities were then correlated with the net activity. Correlations were not significantly different if other time constants from the same study were used.

For linear regression analysis, rotational velocities were separated into right and left rotations by taking the absolute values of all negative or, respectively, the positive rotational velocities. The rotational velocities and forward velocity were fit to determine regression coefficients for Z-scored locomotor parameters (predictors: $v_{Rot\ right}$, $v_{Rot\ left}$, $v_{For}$), using the $\Delta F/F$ signal of a given ROI (left or right nodulus) as observation. Bin size for all parameters was 25 ms. Time shifts (−375 to 750 ms in 25 ms steps) were introduced between observation and predictor variables to get a sense of neural responses over time.

We should note that the GAL4 driver line that we used for these experiments, R37F06, also contains $PB_{G2-9}$.s-FB$l$3.b-$NO_2$V.b (PFN$_V$) neurons (*Wolff et al., 2015*). To avoid potential contamination from this population, we only imaged the top plane of the noduli that contain the P-EN arbors in the $NO_1$ subdivision. $NO_2$, where the PFN$_V$s arborize, is distinct from and more ventral to $NO_1$. The expression of the PFN$_V$ neurons was often too weak to see, but, when visible, $NO_1$ and $NO_2$ could be clearly distinguished.

## Fly preparation for electrophysiology during walking

The fly was anaesthetized on ice and transferred to a cold plate at 4°C. The fly's proboscis was pressed onto its head and immobilized with wax. The fly's head and thorax were UV-glued to a plastic shim, the chamber was sealed with grease, the head covered with saline, and the head capsule opened. The saline used for electrophysiology was the same as used for imaging, except that the concentration of $CaCl_2$ was 1.5 mM instead of 2.5 mM. The fly was positioned on an air-supported

ball and the walking velocity of the fly was monitored using a camera system (*Seelig and Jayaraman, 2015*). For all electrophysiology experiments, we used an 8 mm diameter, 47 mg polyurethane foam ball. Ball movement was recorded at a sampling rate of 4 kHz. Flies were either recorded in darkness, or given closed loop control over a 15° wide blue vertical stripe, presented on a cylindrical LED display spanning 240° in azimuth and 120° in elevation (*Reiser and Dickinson, 2008*). For all experiments presented in this paper, the gain of the closed loop was set to a natural gain of 1, meaning that a 90° rotation of the ball was mirrored as a 90° counter rotation of the stripe in the visual arena. The gap of the arena (in the back of the fly) was ignored, i.e. the stripe ran smoothly from one side of the gap to the next as if the space in between did not exist. For analysis, stripe positions on the 240° arena were mapped linearly to an arena spanning 360°.

## Electrophysiology recordings

For electrophysiology experiments, GFP was expressed in P-EN neurons and visually guided single cell recordings were performed under epifluorescence and IR illumination. Whole-cell and loose patch recordings from P-EN neurons were performed with a recording solution of the following composition (in mM): K-aspartate (140), HEPES (10), KCl (2), EGTA (1.1), CaCl2 (0.1), MgATP (4), Na3GTP 0.5), phosphocreatine (5), and biocytin (0 to 6). The brain was perfused with oxygenated saline containing 1.5 mM CaCl$_2$. The faithfulness of spike rates reported with whole-cell patch-clamp recordings was verified by extracellular loose patch recordings (*Supplementary file 1*). Pipette resistance and capacitance was compensated. For whole-cell recordings, recording quality was monitored throughout the experiment through positive and negative current injections. Access resistance and liquid junction potential was not compensated for. Data was acquired at 20 kHz and low-pass filtered at 4 kHz. Cells were filled with Alexa dyes and neuron identity and soma location was confirmed by epifluorescence imaging and/or post-hoc confocal imaging of PFA-fixated brains.

## Electrophysiological characterization of P-ENs

The input resistance $R_{in}$ was estimated from fitting an exponential decay to the membrane potential after the offset of current injections. Since the offset of a hyperpolarizing current pulse often resulted in rebound spiking and a concomitant overestimation of $R_{in}$ we fitted the offset of depolarizing current injections to derive the passive electrical properties. P-ENs are unipolar neurons with a small soma (diameter <5 μm), a thin primary neurite, and a high input resistance (1.9 ± 0.8 GΩ) close to typical patch clamp seal resistances. Since in that scenario the two resistances function as a voltage divider, the measured membrane potential might be erroneous. We thus performed loose patch recordings from P-ENs to determine their firing rates in an unperturbed recording configuration (3.9 ± 2.6 Hz during standing epochs) and aimed at adjusting the resting membrane potential in whole-cell recordings to a value that would roughly yield similar spike rates. The bias current injected to that end was on average −5.8 ± 3.6 pA. See *Supplementary file 1* for quantification of electrophysiological properties. Patch-clamp recordings of cells with lower input resistance, higher membrane potential, and/or higher bias currents than those reported in *Supplementary file 1* were excluded from analysis, as were cells in which the access resistance was too high to allow the unambiguous detection of action potentials. Both whole-cell and loose patch recordings were terminated (and/or late trials excluded post-hoc) if the spike frequency increased (or decreased) notably, whether it was because of unfavorable changes in signal-to-noise ratio, sudden depolarizations in the whole-cell configuration, or partial break-ins in loose patch recordings. All reported values for the cellular membrane potential are not corrected for liquid junction potential.

## Post-processing of brains and confocal imaging

After the end of a patch-clamp experiment flies were carefully detached from the holder and their brains dissected out. After 10 min fixation in 4% PFA in 0.1M phosphate buffer at room temperature, brains were washed three times for five minutes each in phosphate buffer and mounted in Vectashield on a microscopy slide. Samples were imaged on a Zeiss 710 confocal microscope with a 63x objective. Since P-EN arborizations span the entire dorso-ventral axis of the fly brain, a given set of frames encompassing an individual neuropile was merged in a maximum z-projection and adjusted for brightness and contrast. These individual neuropile projections were then max-projected to yield the overview image shown in *Figure 3A*.

## Spike detection

For both loose patch as well as whole-cell recordings, the recorded voltage was bandpass filtered (50–1000 Hz) and a threshold was set for automatic spike detection. The threshold was set individually for each recording and, if necessary, manually adjusted for individual sweeps (10–300 s). Only recordings with a reasonable signal to noise ratio and sufficient robustness to the setting of the spike threshold were included in the final dataset (see also *Figure 4—figure supplement 2*).

## Characterization of P-EN responses to locomotor parameters

The fly's velocities were calculated from the displacement of the ball as reported by the treadmill (*Seelig et al., 2010*).

To quantify the neural response to rotational velocity, we fit sigmoidal tuning curves to membrane potential and spike rates as a function of rotational velocity. All parameters were binned into 50 ms windows. The membrane potential was 50 Hz lowpass filtered and spike rates were smoothed with a 1 s Gaussian window with a half-width of 0.35 s. After shifting the neural responses by the cell's respective response lag (see below) to align them with the rotational velocities they correlated most strongly with, we excluded all periods in which the fly was not walking. Subsequently, mean values were calculated for 12°/s wide rotational velocity bins and fit with a sigmoid of the form

$$f(v_{rot}) = \frac{r_{max}}{1 + e^{(s*(v_{rot}+v_o))}} + r_{min}$$

with $r_{max}$ being the neurons maximum response, $s$ the slope of the tuning curve, $v_{rot}$ the rotational velocity, $v_0$ the lateral displacement of the curve on the x-axis and $r_{min}$ the minimum response of the neuron. The number of samples that contributed to each mean were used as weights.

Heat maps for spike rates and membrane potential as a function of rotational and forward velocity were constructed from non-overlapping 20 ms bins of data, similar to (*Guo and Ritzmann, 2013*) except that velocities were not smoothed. Spike rates were smoothed with a 1 s Gaussian window with a half-width of 0.3 s and the membrane potential was 50 Hz low-pass filtered. For each time bin, values were assigned to a field in a two-dimensional array depending on the fly's rotational velocity (in a 12°/s grid) and forward velocity (in a 1 mm/s grid). The heat map represents the mean value of each of those fields containing at least 50 time bins (1 s of data).

Since P-EN neurons tend to spike at rest, spike-triggered averages (STAs) of rotational velocities were constructed from only those spikes that occurred during 2 s epochs with rotational velocities exceeding 40°/s at any point in time.

For linear regression analyses, the membrane potential was 50 Hz low-pass filtered. Rotational velocities were separated into right and left rotations by taking the absolute values of all negative or, respectively, positive rotational velocities. The rotational velocities, membrane voltage and spike counts were subsequently binned in non-overlapping 20 ms windows. Generalized linear model regressions were fit to determine regression coefficients for Z-scored locomotor parameters (predictors: $v_{rot\ right}$, $v_{rot\ left}$, $v_{rot\ right}^3$, $v_{rot\ left}^3$), using either membrane potential or spike count as observations. We assumed a normal distribution for the membrane potential (making the procedure an ordinary linear regression) and a Poisson distribution for the spike count (modeling the logarithm of the spike count as a linear combination of the predictors). Time shifts (−2 to 2s in 20 ms steps) were introduced between observation and predictor variables to reveal possible delays between the neural response and the fly's behavior (or vice versa). To calculate the autocorrelation of the rotational velocity, the same method was applied, using $v_{rot}$ as predictor and observation both. In all cells, the regression coefficients for rotational velocity with neural responses followed a stereotypical time course. The neural response lag was defined as the time point of the maximum absolute difference of regression coefficients for ipsi- and contralateral rotations. Inclusion of forward velocity in the generalized linear regression did not yield consistent results across P-ENs. On average, coefficients at zero temporal lag were not different from zero (0.00 ± 0.08, one-sample t-test: p=0.99). Thus, we did not include forward velocity as a parameter for determining temporal properties of P-EN firing.

## Generation of heat maps and tuning curves to the fly's heading

Heat maps for spike rates and membrane potential as a function of rotational velocity and the fly's heading were constructed from non-overlapping 25 ms bins of data. In visual closed loop

experiments, the heading angle was approximated as 2π minus the azimuthal position of the stripe. In experiments where the fly walked in darkness, the heading angle was calculated as the accumulated rotation of the fly. Spike rates were smoothed with a 1 s Gaussian window with a half-width of 0.3 s and the membrane potential was 50 Hz low-pass filtered. For each time bin, values were assigned to a field in a two-dimensional array depending on the fly's rotational velocity (in a 12°/s grid) and the fly's heading (in a π/4 grid). The heat map represents the mean value of each of those fields containing at least eight time bins (0.2 s of data).

For fitting tuning curves to the fly's rotations and virtual heading, we restricted the analysis – for each cell individually – to the range of rotational velocities that was well sampled across all virtual heading angles to avoid strong biases in the sampling of particular heading angles at a given rotational velocity. To that end, we visually inspected grayscale heat maps of a sample size matrix analogously generated to the neural response matrix (see *Figure 5—figure supplement 1B*) and excluded all columns outside a well-sampled rectangle. Rotational velocity tuning curves were fit with a sigmoidal function as outlined above, except that rotational velocity bins were 12.5°/s wide (to fit with the margins of the heading $\bigcup$ rotational velocity array created for the heat map on the basis of which the velocities to be included were determined). Heading angles were binned into 32 bins of 11.25° width. Tuning to the fly's heading angle θ was fit with a compound von Mises function of the form

$$f(\Theta) = r_{min} + \frac{g_1 e^{k_1 \cos(\Theta - \Theta_{0_1})}}{2\pi I_0(k_1)} + \frac{g_2 e^{k_2 \cos(\Theta - \Theta_{0_2})}}{2\pi I_0(k_2)}$$

with $r_{min}$ being the minimum response of the neuron, $g$ the rate modulation of the neuron, $\theta_0$ the preferred heading angle, $\kappa$ the concentration of the modulation around $\theta_0$, and $I_0(\kappa)$ the modified Bessel function of order 0. P-EN heading tuning often showed a peak and a trough (see *Figure 5C*), a phenomenon that was more pronounced on the level of the membrane potential, and was not well described by a unimodal von Mises function. Initial values for the parameters $\theta_{01}$ and $\theta_{02}$ were chosen to be angles of the peak and trough of the smoothed mean spike rates curves.

## Calculation of tuning indices

A rotation tuning and a heading tuning index was calculated for each cell. The rotation tuning index was defined as the $R^2$ value of the sample-size weighted spike rate fit to a sigmoidal function (see above).

The heading tuning index H was defined as the mean vector length of spike rates over all virtual heading angles and calculated according to the following formula:

$$H = \frac{\left| \sum_{n=1}^{N} r_n * e^{i * \theta_n} \right|}{\sum_{n=1}^{N} r_n}$$

with N being the number of heading bins (N = 32), $\theta_n$ the bin center, and $r_n$ the mean spike rate in a given bin. The preferred heading angle was defined as the argument of H, i.e. the phase of the mean vector.

Chance-level statistics for rotation and head direction tuning indices of experiment averages were computed for each cell individually through a shuffling procedure (performed for 5/6 visual closed loop cells, since the 6[th] fly did not sufficiently sample all heading quadrants for a wide enough range of rotational velocities). For each of the 1000 shuffling repetitions, a cell's spike train was chopped into 1s fragments that were randomly re-assembled. The shuffled instance of the two tuning indices was calculated using the scrambled spike train. The significance level was set at the 95[th] percentile of the shuffled rotation and heading indices.

## Epoch identification, conditional tuning curves, conditional tuning indices

Experimental epochs for analysis of heading tuning stability and extraction of conditional tuning curves were identified as follows: All recordings were partitioned in five minute trials for which two-dimensional sample size matrices were constructed as described above (see *Figure 5—figure supplement 1B*) and visually inspected for non-zero bin counts within a rectangular portion of this

matrix that spanned all heading angles and at least 250°/s width of rotational velocities. Practically, we thus selected for epochs in which the flies rotated vigorously on the ball, visiting all possible heading quadrants by balanced turning about the right and left body axis. In one case, two such five minute trials were concatenated to obtain sufficient sampling. The same shuffling procedure as described above was performed for the experimental epochs, except that trials were now chopped in 0.5 s fragments. 6/7 epochs passed the significance threshold (p-value of the failed epoch: p=0.193).

To generate conditional tuning curves for a neuron's tuning to the fly's heading dependent on rotational velocity, separate tuning curves were fit to spike rate and membrane potential as a function of the fly's heading for all epochs of positive versus negative rotational velocities, respectively. To generate conditional tuning curves for P-EN tuning to rotations, separate tuning curves were fit to the fly's spike rate and membrane potential as a function of the rotational velocity for epochs when the fly's virtual heading was aligned with the neuron's preferred versus non-preferred heading quadrant. The preferred heading quadrant was determined from the unconditional tuning curve containing all data points (see *Figure 5C* for an example) and defined as all bins where the firing rate exceeded the half-maximal positive modulation, that is, the baseline (defined as the median firing rate across bins) plus half the difference between baseline and peak firing rate. The non-preferred quadrant was defined as all heading bins where the firing rate was below the median (see *Figure 5—figure supplement 1D* for visualization). Otherwise, the fitting procedure was the same as outlined above: Data was binned into non-overlapping 25 ms bins of data, spike rates were smoothed with a 1 s Gaussian window with a half-width of 0.3 s, membrane potential was 50 Hz low-pass filtered, rotational velocities were sorted in 12.5°/s bins, heading angles in 11.25° bins.

The calculation of conditional tuning indices was performed in an analogous fashion: The conditional rotation tuning index was defined as the $R^2$ value of the conditional tuning curve described above. The conditional heading tuning index was defined as the mean vector length of spike rates over all virtual heading angles, computed separately for rotations to the P-EN's preferred versus non-preferred turn direction.

For quantifications presented in *Figure 6E*, rotational tuning was defined as the difference of the fitted neural response at −150 and 150°/s. Heading tuning was defined as the difference of the conditional tuning curves at the positions corresponding to the maximum and minimum of the unconditional tuning curve that was fitted to all data points irrespective of turn direction. To avoid blurring the boundaries of preferred and non-preferred heading quadrants over time, the analysis of conditional tuning curves presented in *Figure 6* was performed on the 5–10 min experimental epochs presented in *Figure 5—figure supplement 1C*. In one case where two separate epochs had been identified for the same cell, only the first one was included in the analysis. For the analysis of the relative differences in membrane potential versus spike rate modulation presented in *Figure 6—figure supplement 1*, we used these same epochs from all whole cell recordings (at most one per cell) plus the experiment averages of the two neurons that showed significant heading tuning over the course of the experiment, but no sufficient sampling during short epochs.

## Immunohistochemistry, clearing, mounting

Immunohistochemistry was performed largely as described in (*Aso et al., 2014*), but with several exceptions. Standard dissection, fixation and rinse conditions were used, as outlined previously. The concentrations of the two primary antibodies were changed as follows: Rat a-FLAG was diluted at 1:100 and Rabbit a-HA at 1:600. In addition, the tissue was incubated in all three primary antibodies (the two noted above plus mouse α-bruchpilot) simultaneously, diluted in 5% goat serum (GS) in PBT 0.5% Triton X-100 in PBS at RT for four hours on a rotator and then for 48 hr at 4°C on a rotator. Following rinses, as outlined in (*Aso et al., 2014*), tissue was incubated simultaneously in secondary antibodies ATTO647N goat a-rat IgG (1:150; H and L; Rockland # 612-156-120) and CY3 goat a-rabbit (1:1000; Jackson Immuno Research # 111-165-144) for 4 hr at RT and then 2–3 days at 4°, both incubations on a rotator. Following immunohistochemistry, tissue was post-fixed and rinsed, then mounted on coverslips for further processing. Tissue was dehydrated through an ethanol series, then rinsed in xylene before embedding in DPX mounting medium (Electron Microscopy Sciences, Hatfield, PA). Embedded tissue was dried for two days before imaging.

## Microscopy and image analysis

Images were acquired using a Zeiss LSM 710 confocal microscope and a Plan-Apochromat 63x/NA1.4 oil immersion objective. Frame sizes were 1024 × 1024 pixels with a voxel size of 0.19 x 0.19 × 0.38 micrometers, a zoom factor of 0.8 and one frame average. The reference intensity was ramped throughout the depth of a given sample (reference channel not shown here), but the gain and power for the two data channels (synaptic and membrane) were maintained at constant levels within each sample.

Confocal stacks were viewed and analyzed in the image-viewing software 'Janelia Workstation' (S Murphy; K Rokicki; C Bruns; Y Yu; L Foster; E Trautman; D Olbris; T Wolff; A Nern; Y Aso; N Clack; P Davies; S Kravitz; T Safford, unpublished). Since the red channel (synaptotagmin) was weak, the signal was artificially yet uniformly enhanced in the Janelia Workstation so as to be able to see staining in the ellipsoid body, noduli and the protocerebral bridge: this was done only for the P-EN neurons. Additional image processing was not necessary for the E-PG image. The membrane staining (blue) was robust and this channel was not enhanced in the images shown.

As the R37F06 line weakly expresses $PB_{G2\text{-}9}.s\text{-}FBl3.b\text{-}NO_2V.b$ neurons in addition to the P-EN neurons, we used a second GAL4 line, VT008135, for the synaptotagmin experiments. As confirmed by multicolor stochastic labeling images (data not shown), the only protocerebral bridge neurons targeted by this second line are the P-EN neurons. In the three P-EN brains that did not show obviously higher HA-tagged synaptotagmin in the ellipsoid body, expression was relatively weak in the ellipsoid body but still reasonably strong in the noduli. Over all of the brains, staining in the ellipsoid body was not consistently uniform: in most brains, only some tiles of the ellipsoid body were strongly stained.

## Functional connectivity

The brain and VNC were taken out of 5 to 8 days old female flies and laid on a poly-Lysine coated coverslip. The dissection was realized using the minimum level of illumination possible to avoid spurious activation of CsChrimson. Brains were then continuously perfused in a saline solution with 2 mM $CaCl_2$, bubbled with carbogen (95% $O_2$, 5% $CO_2$) at 60 mL/hour throughout the experiment. Imaging was done on a two-photon scanning microscope (Prairie Technologies, Bruker). The excitation wavelength was 920 nm. P-EN neurons were imaged in the protocerebral bridge and noduli. E-PG neurons were imaged in the gall.

CsChrimson was excited with trains of 2 ms, 590 nm light pulses via an LED shining through the objective. Trains were delivered at 30 Hz. The number of pulses was varied between 1, 5, 10, 20 and 30. Only data obtained from 30-pulse stimulations are shown. Each experimental run consisted of 4 repeats, each approximately 20 s long. Runs were themselves repeated every 2 min (approximately). When present, responses did not desensitize.

## Calculation of fluorescence changes over ROIs

For two-color calcium imaging, each volume was collapsed into one maximum intensity image. Regions of interest (ROIs) were then defined over the average of the maximum intensity stacks. For the noduli, ellipses were manually drawn over each side. For the ellipsoid body, ellipses were drawn around the circumference of the P-EN or E-PG neurons and equi-angularly segmented into 16 regions. For the protocerebral bridge, the red and green channels were combined, and 18 polygonal regions of interest were drawn over the glomeruli. 1024 × 1024 pixel stacks consisting of 25–30 planes, each 2 μm apart, were obtained at ~50 mW of power at the end of each bridge experiment to facilitate glomeruli identification.

The stacks were then spatially smoothed using a two pixel wide Gaussian filter, and the mean fluorescence was calculated over each ROI for each maximum intensity image. The lowest 10% ROI means were computed, and all of the ROI means were then divided by the mean of these lowest values and filtered (Savitsky-Golay, third order polynomial) over 11 time steps to match the time scale of the behavior, thereby creating a final $\Delta F/F$.

$\Delta F/F_0$ was calculated for the functional connectivity as follows. $F_0$ was defined as the average signal before the stimulation. Regions of interest (ROI) were obtained by calculating the average projection of the movie and clustering pixels of the projection by a k-means algorithm between ROI and non-ROI pixels. It's worth noting that the selection method relies only on average intensity and not

activity. This is because we want to use the same detection method for responsive and non-responsive runs. This also relies on selecting fields of view as unambiguously containing the neuron of interest — and only the neuron of interest — during the experiment.

### Red/Green channel separation

To test if the red channel was leaking into the green, or vice versa, the raw fluorescence of the noduli ROIs was compared between the red and green channels in flies that expressed GCaMP6f in the P-ENs and jRGECO1a in the E-PG neurons (*Figure 8—figure supplement 1*). No clear correlation was seen between the two. Additionally, the noduli were clearly visible in the green channel and nonexistent in the red, while the outer ellipsoid body was clear in the E-PG expressing red channel but not in the green. ROIs drawn around the noduli showed large fluorescence changes in the green and small responses in the red, while an ROI drawn in a region that overlapped the ellipsoid body E-PG processes showed large responses in the red and only small fluctuations in the green.

### Population vector average

The population vector average was calculating using the circ_mean function from the Circular Statistics Toolbox (See Data analysis, statistical methods below). For the ellipsoid body, 16 equiangular bins were used, between 0 and $2\pi$, corresponding to the ROIs. For the protocerebral bridge, eight equiangular bins were used, again between 0 and $2\pi$, corresponding to the eight glomeruli in which either the P-EN or the E-PG neurons arborize. The corresponding angle was then multiplied by $8/(2\pi)$ to convert it back into an indicator of the number of the glomerulus.

### Statistics of the P-EN and E-PG protocerebral bridge bumps

For protocerebral bridge analysis, changes in the red and green fluorescence transients across the 18 ROIs were sorted into bins according to the rotational velocity at the time of imaging. The right bridge E-PG peak was then used for aligning both the left bridge E-PG signal and the P-EN signal on both sides. That is, for each frame, we drew ROIs corresponding to the estimated boundaries of the 18 bridge glomeruli (see Calculation of fluorescence changes over ROIs), circularly shifted the right ROI fluorescence transients around the nine rightmost protocerebral bridge glomeruli to place the right protocerebral bridge bump peak at glomerulus 14, and circularly shifted the left E-PG bump and the two P-EN bumps by the same offset around their corresponding protocerebral bridge halves. As the P-ENs only arborize in the outer 16 glomeruli and the E-PGs only arborize in the inner 16 glomeruli, the 'empty' glomeruli were skipped when shifting the $\Delta F/Fs$ for alignment. The population vector average of the mean P-EN and E-PG bump on each side was calculated, and the difference found. We also compared the peaks of the E-PG and P-EN traces, and the mean of the peak and PVA differences for all individual pairs of traces to confirm that the trends were consistent regardless of the statistic that was used. Bump amplitudes and half widths were found for each time point and then averaged.

Further, we confirmed that activity offsets were consistent regardless of which indicator combination labeled the two neural populations, arguing against the lag being merely a result of indicator kinetics (*Figure 8G*). Finally, the R37F06 line also contains PB$_{G2-9}$.s-FB$l$3.b-NO$_2$V.b neurons, and so we also compared offsets using a second GAL4 line, VT008135, which only expresses in the P-EN neurons in the bridge. The results were qualitatively the same and quantitatively similar (*Figure 8—figure supplement 2*).

### Statistics of the P-EN and E-PG ellipsoid body bumps

For the ellipsoid body experiments, the data were binned as described above with the exception that 16 equiangular ROIs were now used. The peak of the E-PG ROI was used for alignment, and the ROIs were circularly shifted to place the peak at position 8 (out of 16). The P-EN ROIs were shifted by the same amount to preserve the relative offset. All statistics were then calculated as described above for the protocerebral bridge. We again compared the two color directions and found that, while clear numerical discrepancies existed, all qualitative trends were the same (*Figure 9—figure supplement 2*).

## Data analysis, statistical methods

We used MATLAB (MathWorks, Inc., Natick, MA) and the Circular Statistics Toolbox (*Berens, 2009*) for data analysis. All errors and error bars shown are standard deviation (SD). No statistical methods were used to predetermine sample size. Reported sample sizes refer to the number of flies throughout. One-way ANOVA was used for statistical comparisons between three or more groups, Student's t-test was used to compare differences from zero or between two sets of paired data. Two-way ANOVA was used to compare different experimental groups across conditions.

## Firing rate model

Our model consists of three populations of neurons corresponding to the E-PG neurons, the P-EN neurons for the left side and the P-EN neurons for the right side. We assume that the E-PG neurons make excitatory synapses onto the P-EN neurons and that the P-EN neurons, in turn, make excitatory synapses onto the E-PG neurons. Finally, we assume global inhibition, a necessary requirement for obtaining localized bumps of activity. A third type of neuron implicitly mediates this interaction from the E-PG neurons to the P-EN neurons. The left and right P-EN populations contain nine neurons each (one neuron per protocerebral bridge glomerulus). Based on anatomical estimates (*Wolff et al., 2015*), the E-PG population contains three neurons per protocerebral bridge glomerulus, and thus a total of 54 neurons (27 neurons per protocerebral bridge side). Note that some uncertainties remain regarding the topology of the circuit around the ventral side of the ellipsoid body and the medial and lateral-most protocerebral bridge glomeruli. The P-EN neurons are thought not to arborize in the most medial protocerebral bridge glomeruli, and the E-PG neurons are thought to skip the outermost protocerebral bridge glomeruli (although there is a class of E-PG neuron, the PB$_{G9}$.b-EB.P.s-ga-t.b, that may help fill this gap [*Wolff et al., 2015*]). Thus, it is still not entirely clear how the recurrent loop is closed all the way around the ring, something that we simplify in our model by assuming E-PG and P-EN neurons for each of the protocerebral bridge glomeruli.

Neural activity in each of the populations was modeled as a firing rate: $E(\theta_i, t)$ for the E-PG neurons connecting to the left protocerebral bridge, $E(\theta_j, t)$ for the E-PG neurons connecting to the right protocerebral bridge, $P_l(\theta_i, t)$ for the left P-EN neurons and $P_r(\theta_i, t)$ for the right P-EN neurons. The firing rates obeyed the following equations:

$$eq:model \quad \tau \frac{dE(\theta_i, t)}{dt} = -E(\theta_i, t) + \left[ \alpha \sum_{N(i)} P_l(\theta_{N(i)}, t) K_l(\theta_i - \theta_{N(i)}) \right]_+$$

$$\tau \frac{dE(\theta_j, t)}{dt} = -E(\theta_j, t) + \left[ \alpha \sum_{N(j)} P_r(\theta_{N(j)}, t) K_r(\theta_j - \theta_{N(j)}) \right]_+$$

$$\tau_p \frac{dP_l(\theta_k, t)}{dt} = -P_l(\theta_k, t) + \left[ \frac{\alpha}{3} \sum_{m \in \mathcal{N}_k^l} E(\theta_m, t) - \frac{\beta}{N} \sum_n E(\theta_n, t) + 1 + v_+ \right]_+$$

$$\tau_p \frac{dP_r(\theta_k, t)}{dt} = -P_r(\theta_k, t) + \left[ \frac{\alpha}{3} \sum_{m \in \mathcal{N}_k^r} E(\theta_m, t) - \frac{\beta}{N} \sum_n E(\theta_n, t) + 1 + v_- \right]_+$$

where $\tau = 80ms$ is the time constant of the E-PG neurons, $\tau_p = 65ms$ is the time constant of the P-EN neurons, $\alpha = 10$, and $\beta = 25$. $\theta_{N(i)}$ is the angular position of the P-EN neurons that project to the E-PG neuron at position $\theta_i$. $N_k^l$ and $N_k^r$ are the sets of indices for the E-PG neurons that share a protocerebral bridge glomerulus with, respectively, the left or right P-EN neuron at position $\theta_k$. $K_l(\theta_i - \theta_{N(i)})$ and $K_r(\theta_i - \theta_{N(j)})$ are kernels that describe overlapping connectivity between the E-PG and P-EN populations in the ellipsoid body: $K_l(\theta) = \frac{f(\theta|0, \kappa)}{2} + f(\theta|35°, \kappa)$ and $K_r(\theta) = \frac{f(\theta|0, \kappa)}{2} + f(\theta|-35°, \kappa)$

where $f(\theta|\Delta, \kappa)$ are von Mises distribution functions with $\kappa = 12$:

$$f(\theta|\Delta, \kappa) = \frac{e^{\kappa \cos(\theta - A)}}{2\pi I_0(K)}$$

The inputs $v_+$ and $v_-$ are rectified velocities: $v_+ = [v]_+$ and $v_- = [-v]_+$.

The connectivity matrix of the circuit, corresponding to the previous equations, is shown in *Figure 10—figure supplement 1A*. The time constants of the dynamics, $\tau$ and $\tau_p$, were chosen to account for the experimentally observed phase shifts in the ellipsoid body and the protocerebral bridge. When a population receives a moving bump as its input, it responds with a spatially shifted moving bump. This phase shift is generally close to $\tau v$, where $\tau$ is the time constant of the neural population and $v$ is the angular velocity of the input. We see that this phase shift is proportional to the angular velocity. Moreover, the angular velocity at which the circuit can no longer track an input velocity and saturates is set by the total time constant and the structural phase shift of the circuit $\triangle$ as follows: $v_{sat} = \triangle/(\tau + \tau_\rho)$. Thus, $\tau + \tau_p$ should be lower than 150 ms for the circuit to track velocity inputs of at least 200°/s. The particular values $\tau$ and $\tau_\rho$ were then chosen to obtain the slopes of the phase shifts as a function of the input velocity (*Figure 10D*).

All simulations were performed with the python programming language, and all code is available at https://github.com/hrouault/ang_veloc_integr (*Rouault, 2017*), with a copy archived at https://github.com/elifesciences-publications/ang_veloc_integr.

## Generation of angular velocity tracks for measuring the simulation diffusion coefficient

The time auto-correlation functions recorded angular velocities were extracted and fitted by exponentials (*Figure 10—figure supplement 2*). From these fits, we found a time auto-correlation of 128 ms and a standard deviation of the angular velocity distribution of 54°/s.

In order to test the model over long times and to check for error scaling, we generated artificial angular tracks according to an Ornstein-Uhlenbeck process:

$$\tau \frac{dv(t)}{dt} = -v(t) + \sqrt{2\tau}\sigma\eta(t)$$

where $\eta(t)$ is a Gaussian white noise of variance 1. We took the following parameters for this process: $\tau$ = 120 ms, $\sigma$ = 50°/s.

## Measurement of the diffusion coefficient

We input to the network the result of the Ornstein-Uhlenbeck process $v(t)$ shown above. The circuit integrates this input, and the bump position, $\theta_b(t)$, is recorded over time. We then compared the angular position of the bump with the integrated velocity:

$$\theta_i(t) = \int_0^t v(t')dt'$$

At time scales on the order of the time scale of the circuit, about 200 ms, the difference $\theta_i(t) - \theta_b(t)$ has a finite variance due to the time delay between the input and the actual bump movement. At longer time scales, the behavior of this is diffusive.

The variance of the difference of angles is estimated from the generated trajectories (see *Figure 10G*) as follows:

$$V = \frac{1}{n_s - 1}\sum_{k=1}^{n_s}\left(\theta_i^k(t) - \theta_b^k(t)\right)^2$$

And the uncertainty can be evaluated as:

$$\langle(\theta_i - \theta_b)^2\rangle = V \pm V\sqrt{\frac{2}{n_s - 1}}$$

We then fit the difference curve by the following equation:

$$\langle(\theta_i(t) - \theta_b(t))^2\rangle = \sigma_0^2 + 2Dt$$

where we initialized the origin on $\theta_b$ so that $\theta_b(0) = \theta_i(0)$.

We found $D$= $1.82\times10^{-3}$ rad²/s and $\sigma_0^2$= $1.27\times10^{-3}$ rad².

## Linearity of the angular integration

We sought to evaluate the linearity of the angular velocity integration for our model (*Figure 10—figure supplement 1B*) as a function of the main parameters $\alpha$ and $\beta$. When varying these parameters, we computed a velocity input-output curve similar to the one displayed in *Figure 10E*. We then computed the slope of this curve for low velocities, $\gamma_{\text{lin}}$, in the linear regime. We also computed the ratio of the input and output velocities at the onset of the saturation, $\gamma_{\text{sat}}$. Our linearity values are expressed as follows:

$$L = \frac{\gamma_{\text{sat}}}{\gamma_{\text{lin}}}$$

By definition, if the integration is perfectly linear, $L$ should be equal to 1. Note that this definition is correct because we are not varying $\tau$, $\tau_p$ or $\Delta$, and hence the value of the bump velocity at saturation does not vary.

## shi$^{\text{TS}}$ experiments

All trials were 2 min long and were performed in the dark. The E-PG neurons were imaged with a wavelength of 930 nm. The line VT025957 was used to image these neurons due to its strong GCaMP expression. The first three trials occurred at room temperature. Heated saline was then perfused across the brain until a temperature of 30°C was reached, as measured by a thermocouple placed near the head capsule. The fly was then shown successive 30 s bouts of clockwise and counterclockwise sine gratings for 5 min to evoke rotational optomotor responses, and the temperature was stabilized at 31°C. We used this 5 min period of turning at the restrictive temperature to accelerate depletion of the pool of P-EN synaptic vesicles. Nine more trials were then performed in the dark. Finally, the perfusion was stopped and the bath was allowed to cool down to 26°C. After waiting a further 10 min, six more trials were performed. The fly's behavior was variable in the recovery experiments, with some flies continuing to run, some not moving at all, and many gradations in between. Further, the bump amplitude did not recover across all flies. We therefore decided not to include this data. Statistics for walking performance during imaging are available as part of *Supplementary file 2*.

Significance testing between 31°C distributions (shown in *Figure 11D*) was performed via bootstrapping. For each pair of datasets, the data of the two pairs were combined. Two new pairs, each with the same number of samples as the original pairs were then created from the combined data via random sampling with replacement. The difference of the means of the two new pairs was then calculated. This procedure was repeated 10,000 times to create a distribution of differences. The difference of the means of the original dataset was then compared to this distribution.

When comparing the change in heading to the change in PVA position for turns, we first defined turns as continuous periods where the rotational speed exceeded 15°/s. We then only considered turns where the PVA strength exceeded a certain threshold. This threshold was defined as follows: for the room temperature cases, where the flies primarily rotate on the ball without much forward walking, we expect the E-PG $Ca^{2+}$ activity to roughly scale with rotational speed. As flies exhibits roughly Gaussian distributions of rotational velocities, we would therefore expect the $Ca^{2+}$ activity to exhibit a half-Gaussian distribution over the course of the experiment. The calculated PVA strength will depend on this activity as follows: At low E-PG activity, the noise of the imaging will dominate the signal, leading to a range of low, but non-zero PVA strengths. During periods of high activity, the signal will dominate, leading to a clear bump and a high PVA strength. Thus, we would expect the PVA strength distribution to be peaked, with the values on the low side of the peak being dictated by noise and the values on the high side of the peak coming from the activity itself. We therefore chose a PVA strength threshold, 0.025, that reliably matched or exceeded the peak of the half-Gaussian distribution seen at higher PVA strengths, attributing values above that to the E-PG activity and values below that to the noise of the system.

Once we had selected turns where the PVA strength exceeded the threshold throughout the turn, we then calculated the change in heading and the change in PVA position over the course of the turn and plotted these values against one another. We fit the data with a linear regression model (using the fitlm function in MATLAB) and extracted the slopes and $R^2$ values.

## Acknowledgements

We thank Yi Sun for advice on two-color calcium imaging, Martin Peek, Igor Negrashov and Bill Biddle for help in designing the fly holder for electrophysiology, Adam Taylor and David Ackerman for WaveSurfer development, Lakshmi Ramasamy for help with LabView programming, Scarlett Coffman, Karen Hibbard, and Todd Laverty for fly husbandry, Geoffrey Meissner and the Janelia FlyLight Project Team for brain dissections, histology and confocal imaging for the presynaptic marker experiments, Ellie Sterne for help searching the VT fly collection, Saba Ali, Jack McCarty, Arlo Sheridan, Claire Peterson, and Stanley Tran for useful discussions about the anatomy and structural connectivity of E-PG and P-EN neurons. We thank Ed Rogers and Michael Reiser for generously sharing unpublished fly lines for perturbation experiments. Sung Soo Kim, Hannah Haberkern, and Chuntao Dan provided experimental advice. Emily Nielson created the animation for *Video 1*. We are grateful to Glenn Turner, Michael Reiser, Eugenia Chiappe, Toshi Hige, Alice Robie, Eyal Gruntman, TJ Florence and members of Vivek's lab for useful discussions, and Arseny Finkelstein, Sandro Romani, and Lorenzo Fontolan for feedback on the manuscript. This work was supported by the Howard Hughes Medical Institute.

## Additional information

### Funding

| Funder | Author |
| --- | --- |
| Howard Hughes Medical Institute | Daniel Turner-Evans<br>Stephanie Wegener<br>Hervé Rouault<br>Romain Franconville<br>Johannes D. Seelig<br>Shaul Druckmann<br>Vivek Jayaraman |

The funders had no role in study design, data collection and interpretation, or the decision to submit the work for publication.

### Author contributions

DT-E, Conceptualization, Data curation, Software, Formal analysis, Investigation, Visualization, Writing—original draft, Writing—review and editing, Performed and analyzed two-color calcium imaging and shibire silencing experiments; SW, Conceptualization, Data curation, Software, Investigation, Visualization, Writing—original draft, Writing—review and editing, Performed and analyzed electrophysiology experiments; HR, Conceptualization, Software, Formal analysis, Writing—original draft, Writing—review and editing, Performed all theoretical and modeling work; RF, Investigation, Writing—review and editing, Performed functional connectivity experiments; TW, Data curation, Writing—review and editing, Managed and analyzed presynaptic marker experiments; JDS, Conceptualization, Methodology, Performed pilot calcium imaging experiments that described angular velocity signals in P-EN neurons in the noduli; SD, Conceptualization, Writing—original draft; VJ, Conceptualization, Supervision, Visualization, Writing—original draft, Writing—review and editing

### Author ORCIDs

Daniel Turner-Evans, http://orcid.org/0000-0002-8020-0170
Stephanie Wegener, http://orcid.org/0000-0002-5809-2222
Hervé Rouault, http://orcid.org/0000-0002-4997-2711
Romain Franconville, http://orcid.org/0000-0002-4440-7297
Vivek Jayaraman, http://orcid.org/0000-0003-3680-7378

## Additional files

**Supplementary files**

• Supplementary file 1. Recording parameters of in vivo electrophysiology. Listed are, for all in vivo electrophysiology experiments, the fly's genotype, experimental condition, time spent walking, mean forward, absolute sideslip, and absolute rotational velocity as well as the range of rotational velocities displayed by the fly while walking. Spike rate and membrane potential ($V_m$) at rest were determined for non-walking periods, maximum spike rates are computed for 50 ms windows. All previous parameters are reported as mean and standard deviation across one-minute windows of the entire experiment. The spike threshold is the membrane potential at the peak of the second derivative of the membrane potential before a spike and was determined for a subset of spikes measured at resting membrane potential. The apparent input resistance was computed from a single exponential fit to the decay phase of a depolarizing current pulse. The bias current is a constant current injected to compensate for the leak through the seal resistance, in some experiments this current was readjusted during the recording.

• Supplementary file 2. Imaging genotypes and walking statistics. Listed are, for all in vivo calcium imaging experiments, the fly's genotype, temperature (in the case of the shi^TS flies), percentage of time spent walking, mean forward, absolute sideslip, and absolute rotational velocity as well as the range of rotational velocities displayed by the fly while walking.

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
