## [Decision Letter]

Thank you for submitting your article "Angular velocity integration in a fly heading circuit" for consideration by *eLife*. Your article has been reviewed by three peer reviewers, one of whom, Alexander Borst (Reviewer #1), is a member of our Board of Reviewing Editors, and the evaluation has been overseen by K VijayRaghavan as the Senior Editor. The following individual involved in review of your submission has agreed to reveal their identity: Andreas V M Herz (Reviewer #2).

The reviewers have discussed the reviews with one another and the Reviewing Editor has drafted this decision to help you prepare a revised submission.

Summary

This is a magnificent paper! Based on a clear-cut hypothesis for the neural mechanism underlying head-direction coding in the ellipsoid body of the fly, Turner-Evans et al. continue their discovery story about the neural network underlying spatial navigation in *Drosophila*. Previously, they reported on a neural population termed 'compass neurons' that tracks the fly's heading orientation in visual surrounds and in darkness. Here, they show a novel cell population that is both pre- (in one neuropil) and postsynaptic (in another neuropil) to the compass neurons and moves the activity bump of the compass neurons in darkness, an elegant mechanism to update the fly's heading representation during turns. The study addresses a timely question and has been carried out with great care. It is technically at the highest level, comprising state-of-the-art genetics, voltage recording, calcium imaging and modeling. Given the unresolved neural mechanisms underlying head-direction signaling in vertebrates, this paper provides an important advance toward resolving that question by focusing on the simpler insect organism. In summary, Turner-Evans et al. present an excellent piece of modern systems neuroscience and provide a very interesting solution to a key problem in spatial navigation.

Major Comments

There is an overwhelming amount of supplementary figures. Some supplementary figures could be merged with the respective main figures. Other supplementary figures do not seem to be entirely necessary. Figure 2—figure supplement 1 could be moved to . Figure 3—figure supplement 1, Figure 3—figure supplement 4, Figure 3—figure supplement 5, Figure 3—figure supplement 6, , Figure 7—figure supplement 4 could be removed and, if not already the case, some of the information from these figures could be mentioned in the main text or Materials and methods section.

A lot of text is used to interpret the anatomy of both E-PG and P-EN neurons in terms of their polarity. Why haven't the authors used a presynaptic marker to show that directly?

As the authors mention, it is counterintuitive that the activity bump of E-PNs lags the E-PG bump in the PCB, but leads it in the EB. Based on the conceptual model and the anatomy it would be expected that both bumps overlap in the PCB. Do the data indicate that the peak of the E-PN bump in the PCB is formed by different E-PN cells than the E-PN bump in the EB, and thus that the location of the bump is different for E-PN dendrites versus axons? This was not entirely clear to me.

Subsection “P-EN spiking activity and membrane potential dynamics encode changes in angular velocity”: It is an interesting finding that the coding bandwidth of P-ENs corresponds with the behavioral bandwidth. Was this expected? Does the bandwidth of heading angles in E-PGs in darkness also correlate with behavioral bandwidth? Is every single glomerulus of the PCB only innervated by one P-EN and one E-PG neuron, or by multiple neurons of the same type? If so, different neurons innervating the same glomerulus could have different bandwidths. This point should be discussed more extensively.

Figure 1: Please show a single E-PG neuron. Also, a connectivity map of the bridge and the ellipsoid body like Figure 6 in Hanesch et al., 1989, or Figure 3 of the authors in their Current Biology article 2016 would be helpful. Along these lines, some words about the intriguing bilateral asymmetry of this structure could be included in the discussion.

For Figure 2 the behavioral trace was convolved with the GCamP6f filter before correlation with the calcium signals in the noduli. In my understanding this was not done for the dual color calcium experiments in Figure 7 and Figure 8. Why not? Maybe this would account for the quantitative differences in the PVA offsets for the different indicator combinations.

Figure 1, Figure 2, and 8 provide partial sketches about the system's connectivity but I missed a comprehensive wiring diagram: Which P glomerulus projects via P-ENs to which of the 8 tile-shaped sectors of the ellipsoid body, and which of the 16 wedge-shaped sectors of the ellipsoid body projects via E-PGs to which glomerulus? In particular, one would like to know how successive tile-shaped sectors overlap – I assume by one wedge, but these important details do not seem to be addressed in the manuscript (the movie touches a bit on these issues but can't provide a good solution).

The authors use the population vector to decode the heading direction from the E-PG activity. It would be great if they could elaborate on that concept and discuss whether the PV is also calculated by the fly (and if so: where) or whether the PV is simply a method for the experimentalist to study E-PG activity (if so: does the animal estimate its heading direction, and if so: how?)

Definition of the model system: the notation, e.g., *ω(Θ_i_,t)* or the *∂_t_ ω*, suggests a system of partial differential equations, in contrast to the finite number of neurons (54) in each cell population. This approach also suggests that each cell population is continuous, i.e. without the notion of wedges or tiles with (more or less) homogeneous populations within each sector. In addition, the model does not distinguish between the larger tiles and smaller wedges – in fact, 54 is not divisible by 8 so this choice remains a bit mysterious. On the other hand, the authors' compact model is appealing. But one would like to see a detailed discussion on how the model relates to the experimental (in particular: anatomical) facts.

Predictions of the model system: it would be nice if the authors could run a simulation with time-dependent velocity input, similar to the experimental situation shown in Figure 1. This would greatly help to appreciate the model's predictive power and could be included as a separate new panel in Figure 8.

Figure 2—figure supplement 1: Why are there negative correlation coefficients for CCW in the left nodulus (and for CW in the right nodulus)? Could this phenomenon hint at a pull-push mechanism, with pull (excitation) in front of the E-PG bump and push (inhibition) behind it?

Interestingly, several features of the P-EN neurons do not fit well with the hypothetic wiring scheme, while other, yet unknown neural elements, still need to be identified in order to fully explain the neural network involved in head direction coding. I suggest addressing these issues more clearly and openly in the discussion, in particular the apparent lack of visual responses in P-EN neurons and the unexpected one-glomerulus shift of P-EN activity following the E-PG activity as these will guide future efforts to unravel the missing elements in the mechanisms of head-direction coding.

Why do the authors use the term "glomerulus" for substructures in the protocerebral bridge that Ito et al., 2014, in a widely accepted intention for a uniform nomenclature of insect brain structures have termed "slice" instead? The term "glomerulus" has originally been used to denote substructures in the antennal lobe/olfactory bulb that are reminiscent in organization to a kidney glomerulus (shell and core). Nothing like this is apparent in the slices of the protocerebral bridge and using the term "glomerulus" instead might cause confusion about its internal organization. By the way, the term has only been used in flies and not in other insect species, which is again highly unfortunate. Therefore, I suggest changing the term "glomerulus" to the widely accepted term "slice".

Subsection “An excitatory loop between P-ENs and E-PGs”. It is highly surprising to see that the P-EN neurons apparently do not respond to visual cues. Unfortunately, only two experiments are provided to support this (Figure 3—figure supplement 2). If this is correct, however, the P-EN neurons cannot receive synaptic input (directly or indirectly) from E-PG neurons in the ellipsoid body, because their signaling as shown by Seelig and Jayaraman (2013, 2015) is dominated by visual over proprioceptive feedback cues. This may be the reason why activation of the E-PG neurons resulted in inconclusive responses in P-EN neurons. I think this discrepancy in the results should be more clearly addressed in the discussion. In other insect species there are multiple sets of P-EN and E-PG neurons (e.g. in locust 3 sets of EPG neurons termed CL1 and 2 sets of PEN neurons termed CL2, partly with opposite polarity). How is it in flies? If the situation is similar, these second or third set, not studied here, will likewise contribute to compass coding, and might be in part responsible for the discrepancies between hypotheses and results.

---

## [Author Response]

*Major Comments*

*There is an overwhelming amount of supplementary figures. Some supplementary figures could be merged with the respective main figures. Other supplementary figures do not seem to be entirely necessary. Figure 2—figure supplement 1 could be moved to Figure 2.Figure 3—figure supplement 1*, *Figure 3—figure supplement 4*, *Figure 3—figure supplement 5*, *Figure 3—figure supplement 6, Figure 6—figure supplement 1, Figure 7—figure supplement 4 could be removed and, if not already the case, some of the information from these figures could be mentioned in the main text or Materials and methods section.*

We have largely followed the reviewers’ suggestions and removed non-essential data plots from the manuscript. To follow the e*Life* recommendations for figure size and complexity, we have further split Figure 3 (into the new Figure 3 and Figure 4) and Figure 4 (into the new Figure 5 and Figure 6). We have removed a total of 10 out of 19 supplementary figures: We have merged Figure 2—figure supplement 1 with Figure 2. We have removed Figure 3—figure supplement 1, merged Figure 3—figure supplement 2 and 3 into the new Figure 4—figure supplement 1 and removed Figures 3—figure supplement 5, Figure 3—figure supplement 6, and Figure 7—figure supplement 4, moving the data to tables in the Materials and methods section where appropriate. We have integrated Figure 5—figure supplement 1 into Figure 5 (now Figure 7). We have simplified Figure 6—figure supplement 1 and Figure 6—figure supplement 3 significantly (now Figure 8—figure supplement 1 and Figure 8—figure supplement 2) and removed Figure 6—figure supplement 2, Figure 6—figure supplement 4, and Figure 7—figure supplement 3.

*A lot of text is used to interpret the anatomy of both E-PG and P-EN neurons in terms of their polarity. Why haven't the authors used a presynaptic marker to show that directly?*

We thank the reviewers for their suggestion. We followed their recommendation and used a presynaptic marker to map the polarity of E-PG and P-EN neurons (Figure 7). We hope that ongoing EM reconstructions of E-PG and P-EN neurons will more firmly establish not just the morphology and polarity of these neurons, but also the connectivity of the circuit.

*As the authors mention, it is counterintuitive that the activity bump of E-PNs lags the E-PG bump in the PCB, but leads it in the EB. Based on the conceptual model and the anatomy it would be expected that both bumps overlap in the PCB. Do the data indicate that the peak of the E-PN bump in the PCB is formed by different E-PN cells than the E-PN bump in the EB, and thus that the location of the bump is different for E-PN dendrites versus axons? This was not entirely clear to me.*

We understand the reviewer’s comments to express the concern “The offset in activity in the protocerebral bridge is counterintuitive based on how we have introduced the anatomy and the conceptual model” and the question “Can this offset be explained by different bump locations in the P-EN axons and dendrites?”

For their concern, the reviewers are correct to point out that the protocerebral bridge bump offset is inconsistent with our simple conceptual model as we have introduced it. We believe that the offset can be explained by a combination of factors, and we have attempted to clarify these factors in the text. To elaborate here:

1) Neural and synaptic time constants within the circuit may lead to an offset of E-PG and P-EN bump positions when the fly is walking. When the bump of activity is moving, the delay in the transfer of heading direction from the E-PG bump to the P-EN bump in the protocerebral bridge will cause the P-EN bump to show delayed movement with respect to the E-PG bump. This lag will manifest as an offset in the PVA of the two bumps, with the P-EN bump lagging the E-PG bump in the protocerebral bridge. This is precisely the effect we see in our model (Figure 9, subsection “P-EN calcium activity in the ellipsoid body leads E-PG activity”).

2) As we mention in the Discussion section, we believe that connections between the E-PG neurons and the P-EN neurons in the protocerebral bridge may be both direct and indirect, and shaped by recurrent inhibition. In addition, our presynaptic markers cannot rule out that there might be additional connections from E-PG neurons to P-EN neurons in the ellipsoid body. These additional direct and indirect connections may increase the lag due to time constants and may also shift the P-EN bump position with respect to the E-PG bump (see the following paragraph for further discussion of this point).

To address the reviewer’s question (whether the protocerebral bridge and ellipsoid body bumps may represent the local activity of distinct P-EN neurons), we performed experiments where we simultaneously imaged E-PG activity in the ellipsoid body and the protocerebral bridge. We found the activity bumps in both neuropil to be consistent with anatomical projections (data not shown). The fact that the E-PG and P-EN bumps do not coincide in the protocerebral bridge has, we believe, to do with a dynamic equilibrium between the two populations. Given the anatomical shift between the E-PG and P-EN populations, we could expect to see a shift in the P-EN activity relative to the E-PG activity in either neuropil, the ellipsoid body or the protocerebral bridge. Our conceptual model assumed a direct transfer of activity in the protocerebral bridge and a resulting shift in the ellipsoid body. This is a “protocerebral-bridge-centric” view. An “ellipsoid-body-centric” view that focuses on information transfer in the ellipsoid body would predict different offsets: In this scenario, the P-EN bump would follow the E-PG bump on one side of the protocerebral bridge and lead it on the other. This would be consistent with our observed offset on the ipsilateral side of the bridge. The small shift in P-EN activity relative to E-PG activity in the ellipsoid body and the larger shift in the bridge could, in sum, equal the anatomical offset.

This does raise the question of why we observe a lag (instead of a lead) in the side of the bridge that is contralateral to the turn direction. It has to be said here that the calcium signal in this half of the protocerebral bridge (which increases with increasing turn velocity, see Figure 8), is somewhat enigmatic to us as a whole. We know from the electrophysiology that spiking activity in the contralateral side of the bridge is suppressed with increasing turn velocities. This is in line with our calcium imaging results in the noduli (Figure 2). In the ellipsoid body, we cannot disambiguate the contra- and ipsilateral P-EN populations, though we assume that we only see Ca^2+^ activity from the ipsilateral neurons. Thus, at this point, we cannot explain the P-EN calcium activity in the turn-contralateral side of the protocerebral bridge, neither with respect to intensity nor bump position. As we speculate in the Discussion section, we suspect that we may be imaging the synaptic inputs to the P-EN neurons in the protocerebral bridge rather than seeing a proxy for their spiking activity, which might explain both the intensity and the position of the contralateral bridge P-EN activity. However, the precise nature of these inputs is, at present, unknown to us.

Overall, we cannot yet estimate the relative contributions of these and perhaps other factors to the observed offsets between P-EN and E-PG bumps in the protocerebral bridge and recognize that this is a limitation of our study.

*Subsection “P-EN spiking activity and membrane potential dynamics encode changes in angular velocity”: It is an interesting finding that the coding bandwidth of P-ENs corresponds with the behavioral bandwidth. Was this expected? Does the bandwidth of heading angles in E-PGs in darkness also correlate with behavioral bandwidth? Is every single glomerulus of the PCB only innervated by one P-EN and one E-PG neuron, or by multiple neurons of the same type? If so, different neurons innervating the same glomerulus could have different bandwidths. This point should be discussed more extensively.*

To clarify: We did not mean to make the point that the coding bandwidth of each individual neuron corresponds with the behavioral bandwidth of turning velocities displayed by the fly in that particular experiment. This correlation is in fact rather weak (R^2^ = 0.51, p = 0.094). The information we intended to convey is that, as a population, P-ENs responded to the full range of rotational velocities displayed by the flies as they walked on the ball. To correct the misrepresentation, we have slightly reworded the respective section in the text and changed the plot from ‘behavioral bandwidth = X versus P-EN bandwidth = Y’ style in Figure 3—figure supplement 2I to a category plot with interconnecting lines (now Figure 4—figure supplement 1).

We have also added the point to the discussion that every single glomerulus of the protocerebral brige is innervated by several P-ENs and that hence the diversity in P-EN response bandwidth could be harnessed by the circuit to integrate a broad range of rotational velocities with high precision.

*Figure 1: Please show a single E-PG neuron. Also, a connectivity map of the bridge and the ellipsoid body like Figure 6 in Hanesch et al., 1989, or Figure 3 of the authors in their Current Biology article 2016 would be helpful. Along these lines, some words about the intriguing bilateral asymmetry of this structure could be included in the discussion.*

We have modified Figure 1 to include examples of single E-PG and P-EN neurons and to feature a connectivity map of the bridge and ellipsoid body for both neuron types.

We have also added discussion relating to the bilateral asymmetry of the central complex.

*For Figure 2 the behavioral trace was convolved with the GCamP6f filter before correlation with the calcium signals in the noduli. In my understanding this was not done for the dual color calcium experiments in Figure 7 and Figure 8. Why not? Maybe this would account for the quantitative differences in the PVA offsets for the different indicator combinations.*

The reviewers are correct to point out that quantitative differences in PVA offsets for different indicator combinations may arise from the different rise and decay kinetics of the red and green indicators we used. Not knowing indicator time constants for P-EN neurons, we wanted to minimize the use of deconvolution to cases in which interpretations would not be significantly affected by possible inaccuracies in the time constants we chose. We thus used the “raw” data when we were making direct comparisons of phase relationships between two populations, and acquired that data using both combinations of calcium indicators (an issue explored in more depth in a GENIE Team publication, [31]). We saw that P-EN activity led E-PG activity regardless of the indicator combination used, albeit more strongly in one direction. We are therefore confident in the existence of an offset between the two populations, but we do not wish to push our interpretation too much further.

To gain some intuition for the magnitude of the offset across velocities, and in response to the reviewers’ question, we did simulate offsets that would be predicted given a variety of different time constants. Time constants for the *Drosophila* larval NMJ were taken from (Dana et al., 2016) (Figure 7—figure supplement 3) and for the dissociated neurons from (Chen et al., 2013) (Figure 1) and (Dana et al., 2016) (Figure 2). We assumed Gaussian activity profiles for the E-PG and P-EN neurons with a FWHM of 90^o^ and a variety of phase offsets as a function of velocity. Figure 12 shows the results for a linearly increasing offset between the E-PG and P-EN bumps with increasing velocity. While we can “fit” the data to the linear profile for a choice of time constants, this involves an arbitrary, if bounded, choice of time constants and an assumed function that relates the offset to the phase difference. We expect that such a choice of time constants would lead to a similarly broad range of “extracted” phase difference functions with a deconvolution procedure, so we would prefer not to include it in the main manuscript.

Author response image 1.Simulating the effects of the Ca^2+^ time constants on the observed bump activity.(**A**) Overview of the procedure for convolving a simulated activity profile with on and off GECI time constants. The bump profiles are assumed to be Gaussians with full width at half maxima of 90^o^. Those profiles are then convolved with the difference between a decaying exponential with time constant τ_on_ and a decaying exponential with time constant τ_off_ to create a final simulated activity profile. (**B**) The phase difference between the simulated P-EN Gaussian bump and the simulated E-PG Gaussian bump was assumed to vary linearly with the rotational velocity. (**C**) The simulated phase difference of the activity bumps after being convolved with a variety of time constants. (Top) The time constants are assumed to be those measured after exciting dissociated mouse neurons with 10 action potentials (Chen et al., 2013) (Dana et al., 2016) (Middle). Time constants chosen to best fit the data. (Bottom) The time constants are assumed to be those measured after exciting neurons in the larval *Drosophila* neuromuscular junction at 5 Hz (Dana et al., 2016). For the left columns, the P-EN neurons are assumed to express the green GECI, GCaMP6f, while the E-PG neurons are assumed to express the red GEC, jRGECO1a. The opposite colors are assumed to have been used at right.**DOI:**
http://dx.doi.org/10.7554/eLife.23496.028

*Figure 1, Figure 2 and Figure 8 provide partial sketches about the system's connectivity but I missed a comprehensive wiring diagram: Which P glomerulus projects via P-ENs to which of the 8 tile-shaped sectors of the ellipsoid body, and which of the 16 wedge-shaped sectors of the ellipsoid body projects via E-PGs to which glomerulus? In particular, one would like to know how successive tile-shaped sectors overlap – I assume by one wedge, but these important details do not seem to be addressed in the manuscript (the movie touches a bit on these issues but can't provide a good solution).*

We have added a connectivity map to Figure 1.

*The authors use the population vector to decode the heading direction from the E-PG activity. It would be great if they could elaborate on that concept and discuss whether the PV is also calculated by the fly (and if so: where) or whether the PV is simply a method for the experimentalist to study E-PG activity (if so: does the animal estimate its heading direction, and if so: how?)*

This is an important issue that we are actively exploring, but that we have few definitive answers for at this stage. Nevertheless, we have added a paragraph related to this issue in Discussion section (Paragraph four).

*Definition of the model system: the notation, e.g., ω(Θ_i_,t) or the ∂t ω*,*, suggests a system of partial differential equations, in contrast to the finite number of neurons (54) in each cell population. This approach also suggests that each cell population is continuous, i.e. without the notion of wedges or tiles with (more or less) homogeneous populations within each sector. In addition, the model does not distinguish between the larger tiles and smaller wedges – in fact, 54 is not divisible by 8 so this choice remains a bit mysterious. On the other hand, the authors' compact model is appealing. But one would like to see a detailed discussion on how the model relates to the experimental (in particular: anatomical) facts.*

All the simulations are done in continuous time and the dynamics are written in terms of temporal derivatives (i.e. with∂t operators). In terms of angles, however, everything is indeed discrete, as the reviewer notes. It is then true that the integrals in the equations are actually just sums over all the discrete angles. We chose 54 neurons based partly on known anatomy and partly for simplicity. There are 9 glomeruli in the protocerebral bridge, which have corresponding tiles in ellipsoid body. We thus have 9 tiles that we divide into two wedges and wedges are believed, based on cell counts, to have approximately 3 E-PG neurons (which make 9 * 2 * 3 = 54). For simplification, we consider that there are as many left and right P-EN neurons as E-PG neurons, even though this is not true in the actual fly central complex. Since, at this stage, the precise anatomical interactions between P-EN neurons and E-PG neurons are still uncertain, we chose simplicity over accurate detail. This simple model will be revisited as more anatomical data becomes available. However, we expect that some of the desirable properties of the network (linear angular velocity response, for example) will remain in the more precise models. We agree with the reviewers that this is a very important point, especially for future research and we have now added comments along these lines in the Discussion section.

*Predictions of the model system: it would be nice if the authors could run a simulation with time-dependent velocity input, similar to the experimental situation shown in Figure 1. This would greatly help to appreciate the model's predictive power and could be included as a separate new panel in Figure 8.*

We thank the reviewers for this suggestion, and we have done as the reviewers recommend (see Figure 10). One of the nice features of the dynamics that emerge from the architecture we assumed is the linearity of the angular velocity response (see Figure 10). This means that it should respond well to time varying velocity input. We have checked this in Figure 10 and have measured the diffusion coefficient for the angular integration errors that emerge over the course of these simulations (Figure 10). Overall, we have a system that works well, at least in the absence of noise, for the velocity ranges that are observed experimentally. This is an important point and we have added a description of these properties in the main text.

*Figure 2—figure supplement 1: Why are there negative correlation coefficients for CCW in the left nodulus (and for CW in the right nodulus)? Could this phenomenon hint at a pull-push mechanism, with pull (excitation) in front of the E-PG bump and push (inhibition) behind it?*

The negative correlations in the noduli are consistent with the electrophysiological recordings, where the activity decreases for turns in the contralateral direction. While this suggests a ‘pull – no pull’ mechanism, we see no evidence for a pull-push mechanism.

*Interestingly, several features of the P-EN neurons do not fit well with the hypothetic wiring scheme, while other, yet unknown neural elements, still need to be identified in order to fully explain the neural network involved in head direction coding. I suggest addressing these issues more clearly and openly in the discussion, in particular the apparent lack of visual responses in P-EN neurons and the unexpected one-glomerulus shift of P-EN activity following the E-PG activity as these will guide future efforts to unravel the missing elements in the mechanisms of head-direction coding.*

We have converted the “Other inputs to the compass system” section of the Discussion to “Other elements of the compass system,” and added the following statements:

“For example, we have previously observed that the ring neurons retinotopically respond to visual cues (Seelig and Jayaraman, 2013). As the putative ring neuron axons arborize in the ellipsoid body along with the E-PG dendrites, it may be possible for them to convey visual information to the E-PG neurons, influencing the movement of the bump of activity.”

“Further, we suggested above that the E-PG to P-EN connection in the protocerebral bridge may be indirect and recruit sources of inhibition. There exist a few classes of bridge interneurons, the so-called PBG6-8.sG9.b, PB18.s-GxΔ7Gy.b and PB18.s-9i1i8c.b neurons (Wolff et al., 2015), which may serve as intermediaries in E-PG to P-EN connections. “

And, finally, we have added a mention to a parallel, independent study that we hope to coordinate publication with. This study, from the Maimon lab, finds evidence for a second type of P-EN neuron that may influence bump movement.

We have decided to remove visual response data from the manuscript, for example, the data for closed loop gains of ‘-1’, as these will require a more in-depth exploration to be properly understood.

*Why do the authors use the term "glomerulus" for substructures in the protocerebral bridge that Ito et al., 2014, in a widely accepted intention for a uniform nomenclature of insect brain structures have termed "slice" instead? The term "glomerulus" has originally been used to denote substructures in the antennal lobe/olfactory bulb that are reminiscent in organization to a kidney glomerulus (shell and core). Nothing like this is apparent in the slices of the protocerebral bridge and using the term "glomerulus" instead might cause confusion about its internal organization. By the way, the term has only been used in flies and not in other insect species, which is again highly unfortunate. Therefore, I suggest changing the term "glomerulus" to the widely accepted term "slice".*

We drew heavy inspiration from the anatomical study of Wolff et al., 2015 (and from the earlier study of Hanesch et al., 1989) and would prefer to preserve their nomenclature. We acknowledge that Ito et al., 2014 have termed these structures “slices”, but also note that they make ample use of the term “glomerulus” for regions outside the antennal lobe (specifically, the Bulb and Optic Glomeruli). As a compromise, we mention protocerebral bridge “slices” before we first use the term “glomerulus”. Finally, we did some etymological archeology in response to the reviewers’ concerns about the word itself, and learned that “glomerulus” is derived from the Latin “glomus”, or a “ball of yarn”, which seems to describe the internals of protocerebral bridge slices rather well.

*Subsection “An excitatory loop between P-ENs and E-PGs”. It is highly surprising to see that the P-EN neurons apparently do not respond to visual cues. Unfortunately, only two experiments are provided to support this (Figure 3—figure supplement 2). If this is correct, however, the P-EN neurons cannot receive synaptic input (directly or indirectly) from E-PG neurons in the ellipsoid body, because their signaling as shown by Seelig and Jayaraman (2013, 2015) is dominated by visual over proprioceptive feedback cues. This may be the reason why activation of the E-PG neurons resulted in inconclusive responses in P-EN neurons. I think this discrepancy in the results should be more clearly addressed in the discussion. In other insect species there are multiple sets of P-EN and E-PG neurons (e.g. in locust 3 sets of EPG neurons termed CL1 and 2 sets of PEN neurons termed CL2, partly with opposite polarity). How is it in flies? If the situation is similar, these second or third set, not studied here, will likewise contribute to compass coding, and might be in part responsible for the discrepancies between hypotheses and results.*

The reviewers correctly point out that P-EN angular velocity tuning as assessed by electrophysiology is apparently unaltered in the presence of a stripe that is coupled to the fly’s movements with a gain of 1 or -1. To probe this response further, we performed two color imaging in the protocerebral bridge under three conditions: in the dark, with a stripe with 1x closed loop gain, or with a stripe with a -1x closed loop gain. The results are summarized in Figure 13. As reported previously, E-PG activity follows the stripe/heading for a gain of 1. For a gain of -1, E-PG activity predominantly follows the fly’s heading, with the exception of rare periods where E-PG activity tracks the stripe instead (highlighted in blue). This is also consistent with our previous report, where E-PG activity was often seen to track the fly’s movements rather than visual cue position for a gain of 0.5 (Seelig & Jayaraman 2015, Extended Data Figure 6). Thus, it is not surprising that the P-EN activity is comparable across all three of these conditions, as seen in the electrophysiology and the imaging. We also cannot conclude that “the P-EN neurons cannot receive synaptic input from E-PG neurons” given these results. Visual cues may predominantly drive the E-PG activity under other conditions and, in fact, alternative pathways likely exist to update the E-PG bump position with respect to visual cues, but such conclusions must be left to future experiments. Despite these findings being in accordance with our interpretation and despite the consistency between E-PG and P-EN results, we would prefer to remove the gain = -1 data from the paper as it is an issue that deserves more thorough investigation.

Regarding other types of P-EN/CL2 neurons: although we know of Homberg lab studies that have identified 3 sets of CL1 neurons in locusts, we are unaware of published evidence for two types of CL2 neurons in that system. We are, however, aware of a different *Drosophila* study carried out in parallel with ours that does identify a second class of P-EN neuron. As mentioned earlier, we now briefly refer to this work in Discussion section (with permission from the Maimon lab).

Author response image 2.P-EN and E-PG activity in the protocerebral bridge while the fly controls a 15^o^ wide stripe with a gain of -1.(**A**) Left and right E-PG bump position, as measured by the bump PVA, for a fly viewing a closed loop strip with a gain of 1 (top) and a gain of -1 (bottom) as compared to the position of the visual stripe and to the fly’s heading. (**B**) The mean and standard deviation of the left and right P-EN activity in the protocerebral bridge as a function of rotational velocity when the fly is in the dark or viewing a stipe with a gain of 1 or -1 for one fly. (**C**) Mean (dot) and standard deviation (line) of the mean ipsilatateral P-EN activity (from 0-150 ^o^/s) across flies.**DOI:**
http://dx.doi.org/10.7554/eLife.23496.029